# Dominant negative ATP5F1A variants disrupt oxidative phosphorylation causing neurological disorders

Sara M Fielder [1], Marisa W Friederich [2,3], Daniella H Hock [4,5,6], Jessie R Zhang [1], Liana M Valin [7], Jill A Rosenfeld [8], Kevin T A Booth [9], Natasha J Brown [5,6,10], Rocio Rius [10,11,12], Tanavi Sharma [4], Liana N Semcesen [4], Kim C Worley [8], Lindsay C Burrage [8,13], Kayla Treat [9], Tara Samson [1], Sarah Govert [1], Sara DaCunha [1], Weimin Yuan [1], Jian Chen [14], Jacob Lesinski [1], Hieu Hoang [1], Stephanie A Morrison [1], Farah A Ladha [8], Roxanne A Van Hove [2], Cole R Michel [15], Richard Reisdorph [15], Eric Tycksen [16], Dustin Baldridge [1], Gary A Silverman [1], Claudia Soler-Alfonso [8,13], Erin Conboy [9], Francesco Vetrini [9], Lisa Emrick [7,13], William J Craigen [7,13], Undiagnosed Diseases Network*, Stephen M Sykes [7], David A Stroud [4,5,6], Johan L K Van Hove [2,3,17], Tim Schedl [14,17] & Stephen C Pak [1,17]✉

## Abstract

*ATP5F1A* encodes the α-subunit of complex V of the respiratory chain, which is responsible for mitochondrial ATP synthesis. We describe 6 probands with heterozygous de novo missense *ATP5F1A* variants that presented with developmental delay, intellectual disability, and movement disorders. All variants were located at the contact points between the α- and β-subunits. Functional studies in *C. elegans* revealed that the variants were damaging via a dominant negative genetic mechanism. Biochemical and proteomics studies of proband-derived cells showed a marked reduction in complex V abundance and activity. Mitochondrial physiology studies revealed increased oxygen consumption, yet decreased mitochondrial membrane potential and ATP levels indicative of uncoupled oxidative phosphorylation as a pathophysiologic mechanism. Our findings contrast with the previously reported *ATP5F1A* variant, p.Arg207His, indicating a different pathological mechanism. This study expands the phenotypic and genotypic spectrum of *ATP5F1A*-associated conditions and highlights how functional studies can provide an understanding of the genetic, molecular, and cellular mechanisms of *ATP5F1A* variants of uncertain significance. With 12 heterozygous individuals now reported, *ATP5F1A* is the most frequent nuclear genome cause of complex V deficiency.

**Keywords** ATP5F1A; ATP Synthase; Complex V; Mitochondriopathy; Oxidative Phosphorylation
**Subject Categories** Genetics, Gene Therapy & Genetic Disease; Neuroscience; Organelles

## Introduction

Adenosine triphosphate (ATP) generation and utilization is a highly coordinated and tightly regulated process. Central to this process is mitochondrial ATP synthase, also known as complex V, which is an essential enzyme in cellular metabolism that is responsible for producing ~90% of cellular ATP. ATP synthase is comprised of a central domain that contains a membrane-embedded central ring ($F_o$) organized by repeating c subunits, attached via a central stalk consisting of the δ, γ and ε subunits to a central globular head ($F_1$) (Fig. 1) (Dautant et al, 2018; Galber et al, 2021). Three α- and three β-subunits, encoded by *ATP5F1A* and *ATP5F1B*, respectively, come together to form a hetero-hexameric ring which has the catalytic activity domain, where ATP synthesis occurs. A peripheral arm consisting of the b, d, and F6 subunits

[1]Department of Pediatrics, Division of Newborn Medicine, Washington University in St Louis School of Medicine, St. Louis, MO 63110, USA. [2]Department of Pediatrics, Section of Clinical Genetics and Metabolism, University of Colorado, Aurora, CO 80045, USA. [3]Department of Pathology and Laboratory Medicine, Children's Hospital Colorado, Aurora, CO 80045, USA. [4]Department of Biochemistry and Pharmacology, Bio21 Molecular Science and Biotechnology Institute, The University of Melbourne, Melbourne, VIC 3010, Australia. [5]Murdoch Children's Research Institute, Melbourne, VIC 3052, Australia. [6]Victorian Clinical Genetics Services, Murdoch Children's Research Institute, Melbourne, VIC 3052, Australia. [7]Department of Pediatrics, Division of Hematology & Oncology, Washington University in St Louis School of Medicine, St. Louis, MO 63110, USA. [8]Department of Molecular and Human Genetics, Baylor College of Medicine, Houston, TX, USA. [9]Department of Medical and Molecular Genetics, Indiana University School of Medicine, Indianapolis, IN, USA. [10]Department of Paediatrics, University of Melbourne, Parkville, VIC 3052, Australia. [11]Centre for Population Genomics, Garvan Institute of Medical Research, and University of New South Wales, Sydney, NSW, Australia. [12]Centre for Population Genomics, Murdoch Children's Research Institute, Melbourne, VIC, Australia. [13]Texas Children's Hospital, Houston, TX 77030, USA. [14]Department of Genetics, Washington University in St Louis School of Medicine, St. Louis, MO 63110, USA. [15]Department of Pharmaceutical Sciences, Skaggs School of Pharmacy and Pharmaceutical Sciences, University of Colorado Anschutz Medical Campus, Aurora, CO 80045, USA. [16]McDonnell Genome Institute, Genome Technology Access Center, Washington University in St Louis School of Medicine, St. Louis, MO 63110, USA. [17]These authors jointly supervised this work: Johan L K Van Hove, Tim Schedl, Stephen C Pak. *A list of authors and their affiliations appears at the end of the paper. ✉E-mail: stephen.pak@wustl.edu

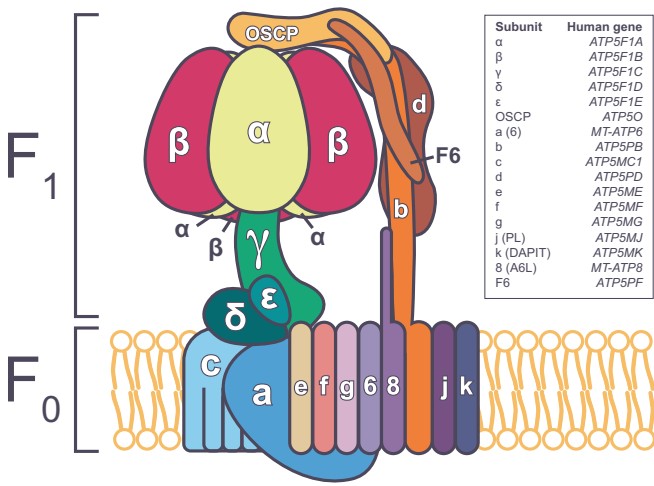

**Figure 1.   A schematic of ATP synthase (Complex V).**

Human mitochondrial ATP synthase is comprised of two domains, $F_1$ and $F_o$. $F_1$ consists of α-, β-, γ-, δ-, and ε-subunits at a ratio of 3:3:1:1:1, and forms the catalytic activity domain where ATP synthesis occurs. $F_o$ forms the proton pore and consists of a membrane-embedded central ring organized by repeating c subunits, and a, e, f, g, j, k, and 8 subunits. A peripheral arm consisting of the b, d, and F6 subunits links to the catalytic head through the ATP5PO (also known as oligomycin sensitivity-conferring protein (OSCP)) subunit, and is embedded in the inner mitochondrial membrane.

| Subunit | Human gene |
| --- | --- |
| α | *ATP5F1A* |
| β | *ATP5F1B* |
| γ | *ATP5F1C* |
| δ | *ATP5F1D* |
| ε | *ATP5F1E* |
| OSCP | *ATP5O* |
| a (6) | *MT-ATP6* |
| b | *ATP5PB* |
| c | *ATP5MC1* |
| d | *ATP5PD* |
| e | *ATP5ME* |
| f | *ATP5MF* |
| g | *ATP5MG* |
| j (PL) | *ATP5MJ* |
| k (DAPIT) | *ATP5MK* |
| 8 (A6L) | *MT-ATP8* |
| F6 | *ATP5PF* |

links to the head through the ATP5PO (also known as oligomycin sensitivity-conferring protein (OSCP)) subunit, and is membrane-embedded with the a, e, f, g, j, k, and 8 subunits. The latter contains the proton pore. The enzyme operates via chemiosmosis which couples the proton gradient generated by the electron transport chain (ETC) with ATP synthesis. Movement of protons from the intermembrane space to the mitochondrial matrix drives the rotation of the central stalk (Boyer, 1997; Kuhlbrandt, 2019; Yoshida et al, 2001). Rotation of the central stalk then induces conformational changes in the α and β subunits of the $F_1$ complex allowing coordinated binding of ADP and inorganic phosphate (Pi) and the catalysis of ATP. One complete rotation of the central stalk produces 3 molecules of ATP (Boyer, 1997).

Genetic variants in *ATP5F1A* associated with human disease have rarely been reported. Two families with biallelic pathogenic variants p.Tyr321Cys and p.Arg329Cys in *ATP5F1A* presented with neonatal or early infantile fatal conditions of devastating neurological lesions, pulmonary hypertension, and cardiac failure (Jonckheere et al, 2013; Lieber et al, 2013). Fibroblasts homozygous for the latter variant showed reduced complex V hydrolytic activity, with decreased amounts of the α, β, and ATP5PO subunits (Lieber et al, 2013). The parents of these children were unaffected. In contrast, two reports described a recurrent de novo heterozygous missense variant p.Arg207His in *ATP5F1A* with life-threatening neonatal-onset mitochondriopathy with failure to thrive, hyperlactatemia, hyperammonemia with elevated glutamine, and decreased citrulline (Lines et al, 2021; Zech et al, 2022). Fibroblasts derived from these individuals displayed reduced ATP5F1A protein, reduced complex V hydrolytic enzyme activity, a lower oxygen consumption rate, and normal inner mitochondrial membrane potential. All three affected individuals showed apparent clinical resolution by 18 months of age with normal development, although

the long-term prognosis is currently unknown. In a second publication, the same p.Arg207His variant was described in a child with transient developmental delay, failure to thrive and lactic acidemia, and reduced complex V activity. Two other dominant de novo variants in *ATP5F1A* p.Arg182Gln and p.Ser346Phe were also reported in individuals who presented with developmental delay, cognitive dysfunction, and dystonia (Zech et al, 2022). Functional studies of these variants were not performed.

Here we describe six probands with heterozygous de novo missense variants in *ATP5F1A*, representing four variants, p.Arg182Gln, p.Ser346Phe, p.Pro331Leu, and p.Leu109Ser. All six probands presented with complex but overlapping neurological phenotypes including developmental delays, intellectual disability, pyramidal tract dysfunction, and dystonia. In vivo functional studies in *C. elegans* provided strong evidence that these variants are damaging to function by a dominant negative mechanism. Furthermore, through protein modeling, proteomics analysis, biochemical and mitochondrial physiology studies using proband-derived cells, we provide new insights into their molecular and cellular mechanism in fibroblasts which are distinct from the previously published dominant (p.Arg207His) and recessive (p.Tyr321Cys and p.Arg329Cys) *ATP5F1A* variants. Our study expands the genetic, clinical, and biochemical description of recurring nuclear complex V defects associated with pathogenic *ATP5F1A* variants.

## Results

### Proband presentations

Proband 1 is a 12-year-old male with dystonia, global developmental delay since birth, and mild to moderate intellectual disability (Table 1). A diagnosis of cerebral palsy was made at three years of age. He crawled up to age three, and then he walked with the aid of braces. His first words were at age 18 months. Neuropsychiatric testing placed him at a 6-year-old level at a chronological age of 10 years. His examination is significant for dystonia, spasticity, gait apraxia, fluency disorder, poor fine and gross motor skills, limited attention, poor muscle tone, and brisk reflexes with normal sensation. He is nondysmorphic. Blood lactic acid levels were mildly elevated. Clinical duo exome sequencing with the father followed by targeted maternal studies identified a heterozygous de novo missense variant in *ATP5F1A* (NM_004046.6:c.545 G > A p.(Arg182Gln)).

Proband 2 is a 9-year-old male with moderate to severe global developmental delay. Early history included significant central hypotonia with crawling achieved at 18 months of age. He can walk with support, but independent ambulation has not been achieved. He has reduced tone with brisk reflexes in upper and lower limbs and upgoing plantar responses, tented upper lip with hypotonic facies and profound central hypotonia. Verbal communication is limited to a handful of words; however, receptive language appears better than expressive language. There has been no regression nor seizures. Growth is in the normal range. Trio genome sequencing identified a de novo heterozygous missense variant in *ATP5F1A* (NM_004046.6:c.545 G > A p.(Arg182Gln)).

Proband 3 is a 3-year-old female with a significant history of severe global developmental delay and abnormal muscle tone,

**Table 1. Clinical characteristics and molecular findings in individuals with ATP5F1A variants.**

| | Proband 1 | Proband 2 | Proband 3 | Proband 4 | Proband 5 | Proband 6 | Individual 7 | Individual 8 | Individual 9 | Individual 10 | Individual 11 | Individual 12 |
|---|---|---|---|---|---|---|---|---|---|---|---|---|
| Reference | This study | this study | This study | This study | This study | This study | Zech et al | Zech et al | Zech et al | Lines et al | Lines et al | Lines et al |
| **Demographics** | | | | | | | | | | | | |
| Age at evaluation | 12 years | 9 years | 3 years | 13 years | 16 years | 11 years | 17 years | 12 years | 14 years[a] | 3 years | 14 months | 18 months |
| Sex | Male | Male | Female | Female | Female | Male | Male | Female | Female | Male | Female | Female |
| **Clinical characteristics** | | | | | | | | | | | | |
| Developmental delay | + | Severe | Severe | Severe | + | Severe | + | + | +[b] | – | – | – |
| Intellectual disability | Mild-moderate | Severe | N/A | Moderate | Severe, nonverbal | Severe | + | Nonverbal | – | – | – | – |
| Progression | – | No regression, very slow motor development | – | Motor progression | Regression | No regression but slow global development | – | – | Resolution | Resolution by 18 months of age | Resolution by 18 months of age | Resolution by 18 months of age |
| Dystonia/spasticity | + | + | + | + | – | + | + | + | – | – | – | – |
| Gait abnormality | + | + | N/A | N/A | + | + | + | + | – | – | + | – |
| Lactic acidosis | + | – | – | – | + | – | – | + | + | + | + | + |
| Ketoacidosis | – | – | – | + | + | – | NR | NR | NR | NR | NR | NR |
| Failure to thrive | – | – | + | + | + | + | NR | NR | + | + | + | + |
| Heart defect | Mildly dilated aortic sinus of valsalva | – | – | – | Atrial septal defect | – | NR | NR | NR | NR | NR | NR |
| **Molecular findings** | | | | | | | | | | | | |
| Protein variant | p.Arg182Gln | p.Arg182Gln | p.Ser346Phe | p.Ser346Phe | p.Pro331Leu | p.Leu109Ser | p.Arg182Gln | p.Ser346Phe | p. Arg207His | p. Arg207His | p. Arg207His | p. Arg207His |
| Coding variant (NM_004046.6) | c.545 G > A | c.545 G > A | c.1037 C > T | c.1037 C > T | c.992 C > T | c.326 T > C | c.545 G > A | c.1037 C > T | c.620 G > A | c.620 G > A | c.620 G > A | c.620 G > A |
| Inheritance | De novo | De novo | De novo | De novo | De novo | De novo | De novo | De novo | De novo | De novo | De novo | De novo |
| Zygosity | Heterozygous | Heterozygous | Heterozygous | Heterozygous | Heterozygous | Heterozygous | Heterozygous | Heterozygous | Heterozygous | Heterozygous | Heterozygous | Heterozygous |
| gnomAD 4.0 | Absent | Absent | Absent | Absent | Absent | Absent | Absent | Absent | Absent | Absent | Absent | Absent |
| CADD score[c] | 33 | 33 | 32 | 32 | 26.7 | 33 | 33 | 32 | 33 | 33 | 33 | 33 |
| REVEL | 0.905 | 0.905 | 0.894 | 0.894 | 0.773 | 0.96 | 0.905 | 0.894 | 0.88 | 0.88 | 0.88 | 0.88 |
| MutationTaster | 0.99 | 0.99 | 0.99 | 0.99 | 0.99 | 0.99 | 0.99 | 0.99 | 0.99 | 0.99 | 0.99 | 0.99 |
| SpliceAI, donor gain, loss | 0.02, 0.00 | 0.02, 0.00 | 0.00, 0.00 | 0.00, 0.00 | 0.00, 0.00 | 0.00, 0.00 | 0.02, 0.00 | 0.00, 0.00 | 0.00, 0.00 | 0.00, 0.00 | 0.00, 0.00 | 0.00, 0.00 |
| SpliceAI, acceptor gain, loss | 0.02, 0.02 | 0.02, 0.02 | 0.00, 0.01 | 0.00, 0.01 | 0.00, 0.01 | 0.00, 0.00 | 0.02, 0.02 | 0.00, 0.01 | 0.03, 0.00 | 0.03, 0.00 | 0.03, 0.00 | 0.03, 0.00 |
| **Functional evaluation** | | | | | | | | | | | | |
| | This study | This study | This study | This study | This study | This study | NR (Zech et al) | NR (Zech et al) | ⇓ ATPase activity (Zech et al) | ⇓ ATP5F1A expression, ⇓ ATPase activity, ⇓ OCR, normal membrane potential (Lines et al) | | |

+ feature present, – feature absent, N/A not applicable, NR data not reported.
[a]Presented during the first few months of life.
[b]Reported as resolved with no lasting neurological problems.
[c]Using CADD GRCh37-v1.7, published 2024.

characterized by central hypotonia and peripheral spasticity. Her medical history is further complicated by dysphagia, chronic constipation, frequent vomiting, and feeding difficulties necessitating the placement of a gastrostomy tube (G-tube). She had a mildly elevated lactate, which at other times was normal. Clinical exome sequencing identified a heterozygous variant of uncertain significance in *ATP5F1A* (NM_004046.6:c.1037 C > T p.(Ser346Phe)). Subsequent parental testing confirmed that the *ATP5F1A* variant in this patient is de novo.

Proband 4 was born prematurely at 33 weeks gestation. At 1 year of age, developmental delays were noted. Over the next years, she gradually developed spasticity and dystonia, and could only walk with adaptive devices. On exam, she has dystonia most notable in facial and upper extremity movements. She has severe spasticity, most pronounced in her lower extremities, and has contractures in elbows and knees. She is fed by gastrostomy tube for dysphagia and has severe gastroesophageal reflux. While non-fasting, she had elevated ketones with 3-hydroxybutyrate 1.96 mM and acetoacetate 0.50 mM, and an elevated ratio 3.89 (normal 0.47-2.57). At age 13 years, she has growth retardation with height at -4 SD and weight at -3.3 SD. Genetic testing identified a de novo heterozygous variant in *ATP5F1A* (NM_004046.6:c.1037 C > T p.(Ser346Phe)).

Proband 5 is a 16-year-old female with significant intellectual disability, developmental regression, growth failure, and history of hospitalizations for ketoacidosis. Prenatally, she was noted to have a dilated right ventricle in her brain, thought to be due to an ischemic event. Neonatally, she had hyperbilirubinemia, which improved with phototherapy. She had failure to thrive, an atrial septal defect, sinus tachycardia with normal QTc, and mild scoliosis. Developmental concerns were raised at 6 months of age. She walked at 21 months with stiffness and toe walking, which has improved. She is currently nonverbal. She has had 8 hospitalizations with ketoacidosis (3-hydroxybutyrate 3.6 mM), typically with normal ammonia, with the first episode at 4 years of age requiring PICU admission. Her ketones normalize when she is healthy. She also has intermittent elevations of blood lactate (1.1–5.5 mM) with normal lactate:pyruvate ratio and urine lactate, without a clinical correlate. Clinical trio exome sequencing identified a de novo heterozygous missense variant in *ATP5F1A* (NM_004046.6: c.992 C > T p.(Pro331Leu)).

Proband 6 is an 11-year-old male with global developmental delay, limited verbal communication, postnatal failure to thrive with feeding difficulties, axial hypotonia, and leg spasticity with contractures (and therefore is nonambulatory). Trio whole genome sequencing identified a de novo heterozygous missense variant in *ATP5F1A* (NM_004046.6:c.326 T > C p.(Leu109Ser)).

All six probands reported here show a shared phenotype of failure to thrive, developmental delay, and a motor syndrome of pyramidal tract involvement and dystonia. Lactic acid has been variable and mildly elevated, but two probands had excessive ketosis. Magnetic Resonance Imaging (MRI) of the basal ganglia/striatum and the brain stem showed no abnormalities (Appendix Fig. S1). A more comprehensive clinical report for each proband is available in the Appendix Full Clinical Report. The phenotypes observed in our six probands are in clear contrast to the phenotypes of the previously reported recurrent p.Arg207His variant, suggesting phenotypic expansion. Moreover, all individuals with p.Arg207His variant had neonatal onset mitochondriopathy that showed clinical resolution in late infancy (Lines et al, 2021; Zech

et al, 2022), whereas the six probands in the current study have phenotypes that are either static or worsening (Table 1).

## Bioinformatic features of *ATP5F1A* variants

Clinical exome or whole genome sequencing identified de novo heterozygous missense variants in *ATP5F1A* (NM_004046.6) as the most likely candidate disease gene for each of the six cases (Table 1). The variants are absent from gnomAD 4.1 and predicted to be deleterious by in silico analysis: (CADD, all >25), pathogenic (REVEL, all >0.7), and disease causing (MutationTaster), while SpliceAI did not predict any effect on splicing (Table 1) (Ioannidis et al, 2016; Jaganathan et al, 2019; Schwarz et al, 2014; Zech et al, 2022). *ATP5F1A* is expected to be tolerant to loss-of-function variants (pLI = 0.29; o/e = 0.3–0.58), thus it is unlikely that haploinsufficiency is a genetic mechanism for *ATP5F1A*-related disease. Conversely, gnomAD data suggests that *ATP5F1A* is likely intolerant to missense variation (Z = 3.16; o/e = 0.63-0.73) (Chen et al, 2024).

## Structural analysis: proband variants are located at the contact points between α:β and α:γ subunits of the F1 complex of ATP synthase

To better understand the mechanism by which the proband variants might impact ATP synthase function, we mapped the locations of the variants to the cryoelectron microscopy-derived structure of the protein (Lai et al, 2023). All four affected residues, L109, R182, P331, and S346 are located at or near the contact points between α:β (ATP5F1A:ATP5F1B) or α:γ (ATP5F1A:ATP5F1C) subunits of the F1 complex (Figs. 2A,B and EV1). The P331 residue lies adjacent to P330 and is located in a pocket near the contact point with the β- and γ-subunits (Fig. 2C). Prolines are often found in turns and loops, and their distinctive cyclic sidechains provide conformational rigidity. While no hydrogen bonds are created or destroyed, the substitution of P331 with leucine dislocates the residue away from the β-subunit and moves it closer to the γ-subunit (Fig. 2D). This displacement, along with a change in flexibility is likely to compromise the structure of this region and alter interactions between the α−, β- and/or γ-subunits.

The S346 residue interacts directly with the β-subunit via a hydrogen bond with M272 (β-subunit). S346 also forms multiple intra-α-subunit hydrogen bonds with E350, F342, and Y343 residues (Fig. 2C). The S346F substitution leads to the loss of hydrogen bonds with Y343 (α-subunit) and M272 (β-subunit) (Fig. 2D). M272 (β-subunit) lies adjacent to the hydrophobic pocket (aa 274–28) of the β-subunit, which is predicted to be critical for the interaction with the γ-subunit (Abrahams et al, 1994; Ganetzky et al, 2022). Therefore, the loss of the hydrogen bond with M272 (β-subunit) likely alters interaction between all three subunits.

Within the α-subunit, the R182 residue forms multiple hydrogen bonds with residues: E350, P177, G178, and I179 (Fig. 2C,D). The R182Q substitution is predicted to result in loss of hydrogen bonds with E350, P177, and G178 (Fig. 2D). The loss of three intra-α-subunit hydrogen bonds will likely have a significant impact on the protein confirmation. Interestingly, both S346 and R182 form hydrogen bonds with a common partner E350 (α-subunit), which

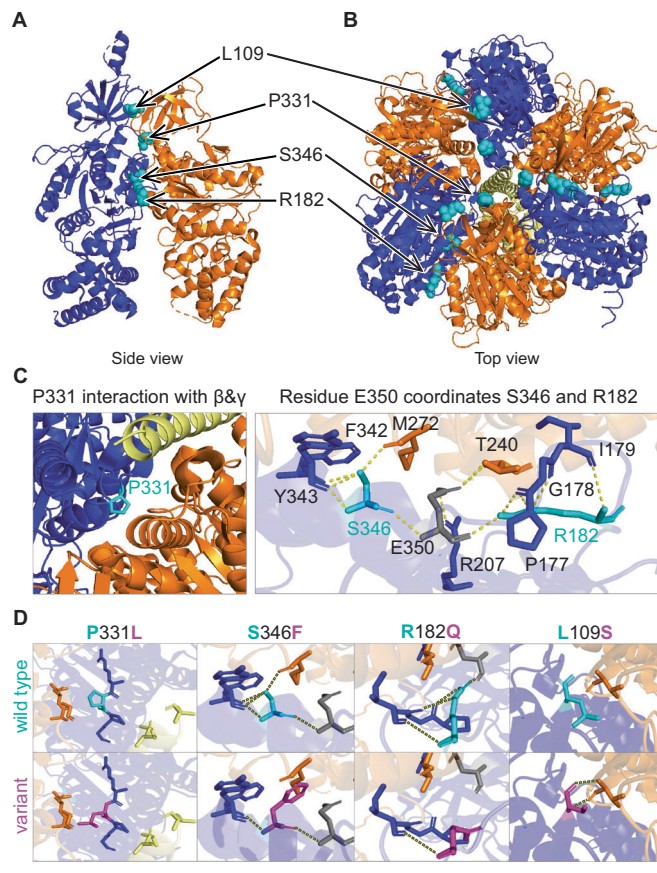

**Figure 2. Proband *ATP5F1A* variants likely disrupt interaction between α, β, and γ subunits.**

Proband variants are located at the interfaces of α:β, and α:γ subunits. Locations of the four variants in a single α−β dimer (**A**), or in the hexameric complex with the γ subunit (**B**). α-subunit is depicted in dark blue, β-subunit in orange, γ-subunit in yellow, and variant locations are shown in cyan. (**C, D**) Close up schematic of wild-type and variant interactions in the holocomplex. Wild-type residues in cyan, proband variant residues in magenta, other residues in α-, β- and γ-subunits are in dark blue, orange, and yellow, respectively. The P331L, S346F and R182Q substitutions are predicted to destroy H-bonds and/or impact the conformation and interaction between α, β, and γ subunits. The L109S substitution creates two novel H-bonds between α, and β subunits which likely alters the interaction between these subunits.

interacts directly with T240 on the β-subunit via a hydrogen bond (Fig. 2C). This interaction with a common partner, E350, suggests that the S346F and R182Q variants act in a similar mechanism by impacting interaction with the β-subunit. The wild-type L109 residue is located at the interface between α- and β-subunits. The substitution of the non-polar, hydrophobic leucine with a polar, hydrophilic serine results in the formation of two new hydrogen bonds between the variant S109 (α-subunit) and V26 in the β-subunit. These changes likely alter the conformation of this region and change the interaction between these subunits (Fig. 2D). We hypothesize that these structural changes caused by the proband substitutions impact ATP synthase function by (1) disrupting proper formation of the ATP synthase complex, (2) impairing ADP + $P_i$ binding, ATP synthesis or release, and/or (3) disrupting interaction between the γ-stalk with α- and β-subunits within the complex and thereby uncoupling the proton motive force from

ATP synthesis. The possible disruption of complex V subunit interactions suggests that the *ATP5F1A* variants may have a change of function genetic mechanism.

## Proband variants are damaging to ATP-1 function in vivo

To determine if the proband variants are damaging in vivo and uncover the genetic mechanism of dominance observed in the probands, we modeled three variants (p.Arg182Gln, p.Pro331Leu, and p.Ser346Phe) in *C. elegans*. The fourth proband variant, p.Leu109Ser, was identified late in the project and thus was not modeled. *ATP5F1A* is highly conserved as *atp-1* in *C. elegans*, with 78% identity and 88% similarity between human and *C. elegans* proteins (Appendix Fig. S2). All three variant residues are conserved in *C. elegans* in regions of high homology (Fig. 3A; Appendix Fig. S2). This degree of conservation enabled us to employ CRISPR-Cas9 genome editing to knock in the proband variants into the corresponding *atp-1* residues (Arg167, Pro316, and Ser331) and evaluate the functional consequences of the missense variants in vivo (Fig. 3B).

*atp-1* deletion (null) homozygotes were first stage larval (L1) lethal, while deletion heterozygotes developed normally, were fertile, and superficially similar to wild-type. Initial attempts to recover viable *atp-1*[R167Q] variant lines following CRISPR-Cas9 editing were unsuccessful as the variant animals were mostly embryonic lethal, even as heterozygotes. To facilitate the study of the R167Q and other proband variants, we first inserted an extra wild-type genomic *atp-1* copy into a safe harbor locus on chromosome IV (Fig. 3C) using the recombination-mediated cassette exchange method (RMCE) as previously described (Marom et al, 2023; Nonet, 2020). We reasoned that transgenic extra wild-type copies of *atp-1* would suppress the damaging effects of the variant alleles and allow them to grow as fertile adults. RNA-seq analysis of homozygous *atp-1*[extra copy] animals showed that they express the transgenic copy at approximately half the level of the endogenous *atp-1* locus on chromosome I (Appendix Fig. S3A). Nevertheless, larval lethality of the homozygous endogenous *atp-1* locus deletion was fully rescued by the two copies of the wild-type transgenic *atp-1* on chromosome IV, allowing the animals to develop into fertile adults (Appendix Fig. S3B). This indicated that the transcriptional activity of the transgenic *atp-1* allele was sufficient to suppress the damaging effects of the variant alleles. Moreover, additional copies of *atp-1* in wild-type animals did not appear to be detrimental as homozygous *atp-1*[extra copy] animals were wild-type in terms of growth rate, body size, and crawling speed, although a minor decrease in thrashing speed was noted (Appendix Fig. S3C–F).

Using the *atp-1*[extra copy] lines, we were able to successfully generate viable animals harboring the proband variants R167Q, P316L, and S331F at the endogenous *atp-1* locus. In total, we generated two independent variant lines and one control line for each variant. The control lines are wild-type at the variant residues, but harbor synonymous changes introduced during variant editing to generate a novel restriction site to aid in allele genotyping and prevent re-cleavage at the edited locus (Appendix Table S1A,B).

We first assessed the effects of *atp-1* variants on growth rate through post-embryonic development. Homozygous deletion (null), P316L, S331F, and R167Q variants without extra copies of *atp-1* died as embryos or arrested at the L1 stage. Therefore, we

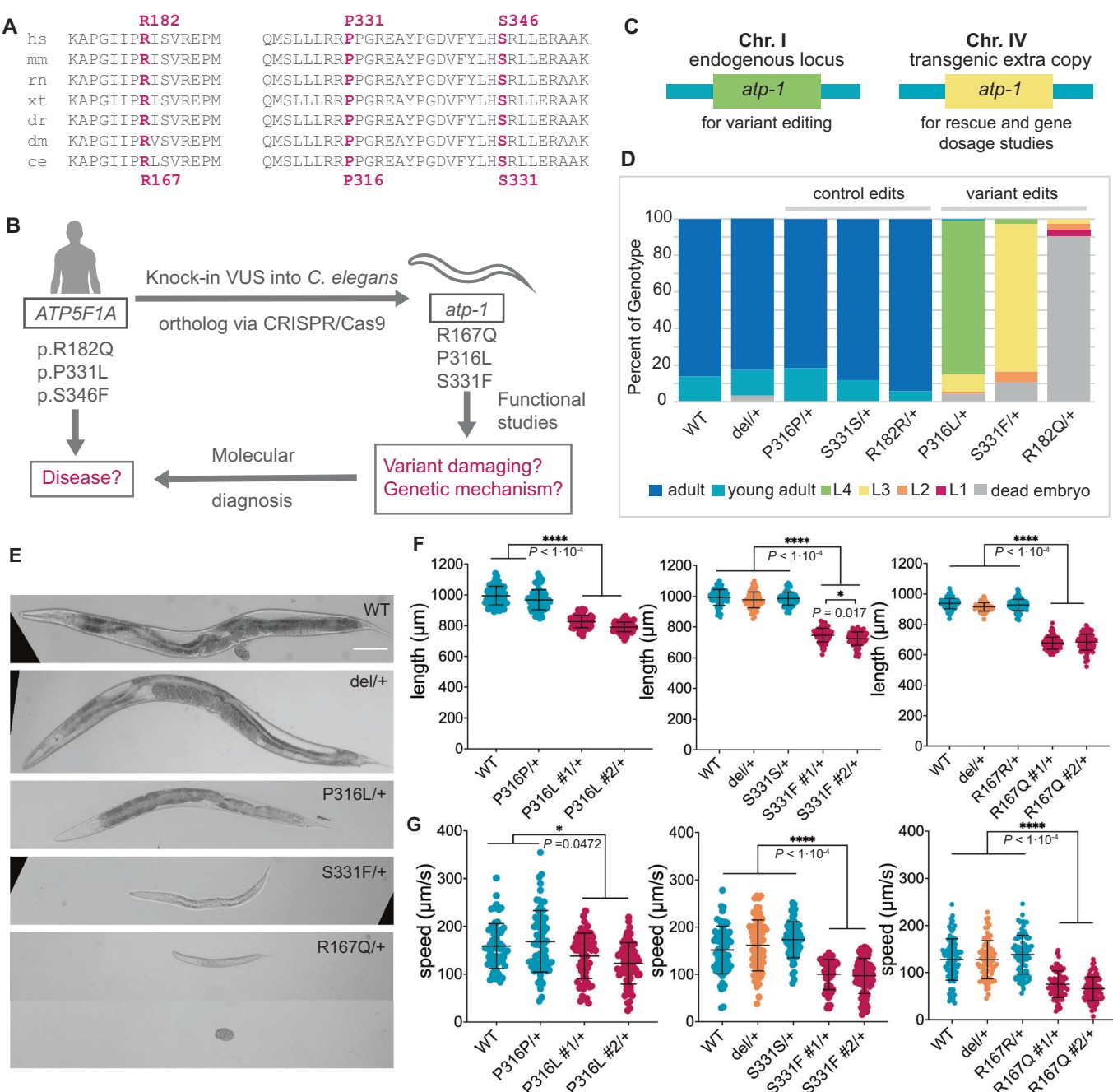

**Figure 3. P316L, S331F, and R167Q variants are damaging to *atp-1* function in *C. elegans*.**

(A) R182, P331, and S346 residues are in areas of high conservation across multiple species, and all three residues are conserved in the *C. elegans* homolog *atp-1*. hs- human, mm- mouse, rn- rat, xt- xenopus, dr- zebrafish, dm- Drosophila, ce- *C. elegans*. (B) Schematic of how variant modeling in *C. elegans* aids in molecular diagnosis of rare disease. (C) Schematic of *atp-1* extra copy lines generated that have endogenous *atp-1* on chromosome I, and a single copy transgene insertion of wild-type *atp-1* on chromosome IV. (D) Development of heterozygous variant animals scored at 72 h after embryo laying. Data from 4 biological replicates are shown. The exact number of animals scored for each genotype is shown in Appendix Table S5. More information on the experiment and the crossing scheme used to generate these animals can be found in the Materials and Methods and in Appendix Fig. S7. (E) Representative images of animals in (D). Scale bar, 100 μm. (F) Length of heterozygous variant animals 24 h post L4 larval stage as measured by WormLab over a two-minute recording. Data for the P316L, S331F and R167Q variants are from 3, 4 and 5 biological replicates, respectively. Each data point represents measurements from a single animal. The exact number of animals analyzed for each genotype is shown in Appendix Table S5. Mean and SD plotted, one-way ANOVA followed by post-hoc Holm–Sidak tests. (G) Crawling speeds of heterozygous variant animals 24 h post L4 larval stage as measured by WormLab. Data for the P316L, S331F and R167Q variants are from 3, 3 and 4 biological replicates, respectively. Each data point represents measurements from a single animal. The exact number of animals analyzed for each genotype is shown in Appendix Table S5. Mean and SD shown. One-way ANOVA followed by post-hoc Holm–Sidak analysis for normal distribution were performed. A Kruskall–Wallis test was performed followed by Dunn's comparison for non-normally distributed data. *$P < 0.05$, ****$P < 0.0001$. Source data are available online for this figure.

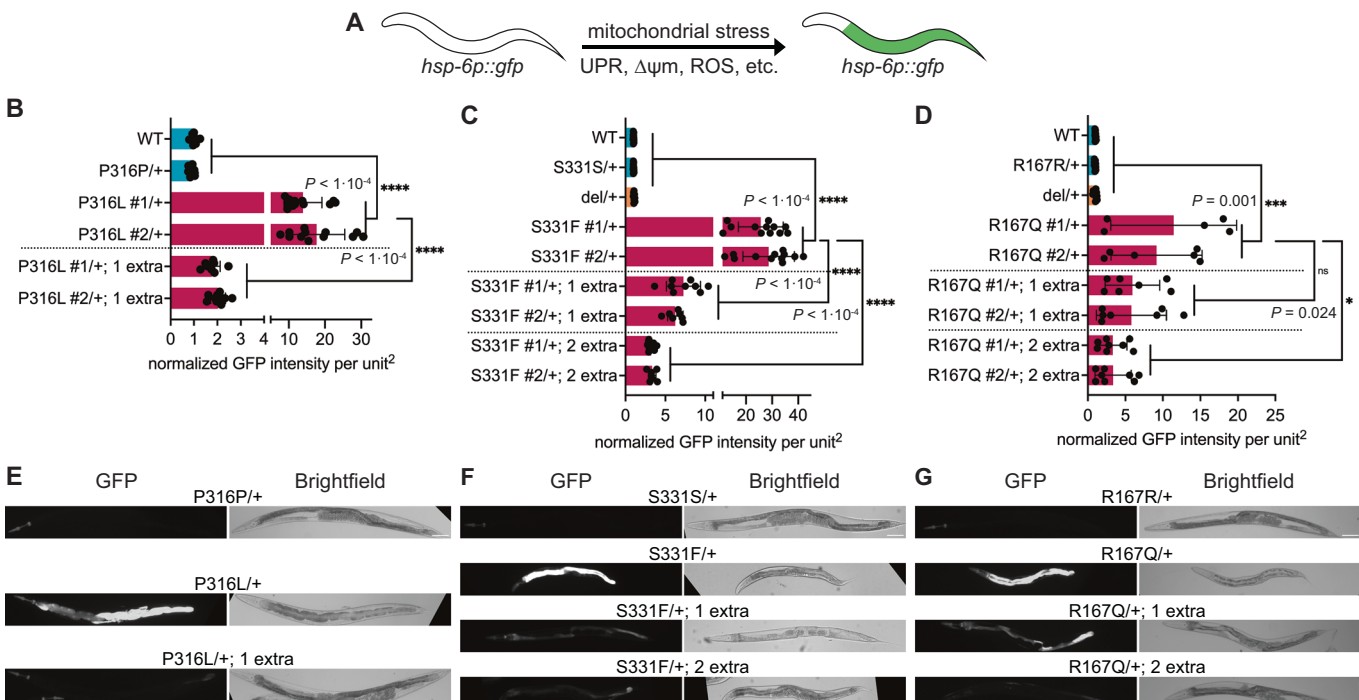

**Figure 4. atp-1 P316L, S331F, and R167Q variants induce mitochondrial stress.**

(A) Schematic of the *hsp-6p::gfp* reporter expressing strain. GFP is not expressed under normal conditions. However, upon mitochondrial stress, the *hsp-6p::gfp* reporter is transcriptionally upregulated in the intestine of the animal. (B–D) Normalized GFP fluorescence expression in 1-day adult animals. Data for (B–D) are from 4, 4 and 3 biological replicates, respectively. Each data point represents measurements from a single well containing 10 animals per well. The exact number of animals analyzed for each genotype is shown in Appendix Table S5. Graphs display mean and SD. One-way ANOVA was performed to determine significance. (E–G) Representative images of animals quantified in (B–D). Scale bar, 100 μm. See Appendix Fig. S7 for crossing scheme to generate these animals. ns not significant, *P < 0.05, ***P < 0.001, ****P < 0.0001. Source data are available online for this figure.

assessed growth rates of heterozygotes. At 72 h post embryo lay, 100% of wild-type and control edit heterozygotes developed into healthy, fertile adults. However, most of the P316L and S331F heterozygotes only grew to L4 and L3 larval stages, respectively (Fig. 3D,E). In contrast, the majority (~90%) of R167Q heterozygotes were embryonic lethal, and the remaining 10% grew to the L1 or L2 stages by 72 h. No growth defects were observed in deletion (null) heterozygotes as 100% grew to mature adults by 72 h, similar to wild-type and control edits (Fig. 3D,E). Together, these results indicated that the P316L, S331F, and R167Q variants are (1) damaging to *atp-1* function, (2) behave dominantly, and (3) are phenotypically different to the deletion (null) allele.

Although the P316L, S331F, and R167Q variant heterozygotes were markedly slow growing, some animals eventually developed into adults. This allows us to compare their body size and crawling speed using WormLab (Roussel et al, 2014). Body lengths of adults heterozygous for P316L, S331F, and R167Q were all significantly smaller (≤800 μm) than wild-type, control edits and deletion heterozygotes (~1000 μm) (Fig. 3F). In addition, all three variant heterozygotes moved at a significantly slower crawling speed compared to their respective controls (Fig. 3G). Similar results were obtained for thrashing cycle (swimming) in liquid (Appendix Fig. S4A–C). Additionally, the absence of phenotypes in deletion heterozygotes indicates that *atp-1* is not a haploinsufficient gene in *C. elegans*. Based on these findings, we concluded that P316L,

S331F, and R167Q are all change-of-function variants, possibly dominant negatives.

## Proband variants cause mitochondrial stress in vivo

Since *ATP5F1A* encodes a subunit of the mitochondrial ATP synthase (complex V), we hypothesized that proband variants altered mitochondrial function. To test this, we utilized a previously described mitochondrial unfolded protein response (UPR^mito) reporter, *hsp-6p::gfp*, to measure mitochondrial stress (Taylor et al, 2014; Yoneda et al, 2004). Under normal conditions, the ATFS-1 transcription factor is imported into mitochondria and degraded. However, under conditions of mitochondrial stress (e.g., low membrane potential, impaired protein import, accumulation of reactive oxygen species, unassembled subunits of the electron transport chain, or unfolded proteins), ATFS-1 does not enter the mitochondria and instead localizes to the nucleus where it facilitates the transcription of mitochondrial stress response factors, such as HSP-6, to help recover mitochondrial function (Fig. 4A) (Anderson and Haynes, 2020; Haynes et al, 2010; Rolland et al, 2019; Taylor et al, 2014; Yoneda et al, 2004). In wild-type, control edit and deletion heterozygotes, *hsp-6p::gfp* expression was barely detectable (Fig. 4B–G). However, in P316L, S331F, and R167Q heterozygotes, *hsp-6p::gfp* expression was strongly upregulated indicating significant mitochondrial stress, supporting the hypothesis that the variants disrupt mitochondrial function (Fig. 4B–G).

## Proband variants act through a dominant negative mechanism

Dominant phenotypes observed with the *atp-1* variants, but not the deletion (null) (Figs. 2 and 3), as well as the dominant presentation in the *ATP5F1A* probands suggest a change-of-function genetic mechanism. From classic gene dosage studies, if the genetic mechanism is antimorphic/dominant negative, then the addition of wild-type transgenic copies should suppress the phenotype (Muller, 1932). The mitochondrial stress phenotype (i.e., induction of *hsp-6p::gfp* expression) was completely suppressed by the addition of one extra copy of wild-type *atp-1* for P316L (Fig. 4B,E). Addition of one extra wild-type copy of *atp-1* partially suppressed mitochondrial stress for S331F and R167Q, and addition of two extra copies further suppressed mitochondrial stress to close to wild-type levels (Fig. 4C,D,F,G). The absence of a phenotype in deletion (null) heterozygotes and the suppression of mitochondrial stress in variant heterozygotes by the addition of extra copies of the wild-type allele provides evidence that the P316L, S331F, and R167Q variants act as dominant negative alleles.

## Proband variant alters mitochondrial morphology in vivo

The dynamics of mitochondrial fission/fusion can change in animals undergoing mitochondrial stress to help maintain ATP homeostasis (reviewed in (Liu et al, 2020) and (Campbell and Zuryn, 2024)). For example, during *C. elegans* starvation, mitochondria are known to fuse to help produce more ATP (Gomes et al, 2011; Liesa and Shirihai, 2013; Rambold et al, 2011; Tondera et al, 2009; Weir et al, 2017). Conversely, under nutrient-rich conditions, mitochondria undergo fission in an effort to decrease ATP production (Molina et al, 2009). To evaluate the effect of the proband variant on mitochondrial morphology, a transgene expressing a mitochondrial-targeted GFP (mito-GFP) was used to label the mitochondria in the body wall muscles (Fig. 5A) (Rolland et al, 2013). High-resolution images were acquired by confocal microscopy, and image analysis was performed using Mitochondrial Segmentation Network (MitoSeg-Net), a pretrained deep learning segmentation model for quantifying mitochondrial morphology (Fischer et al, 2020). Since all three variants induced mitochondrial stress, we performed these studies only in the P316L variant. In wild-type and P316P control muscle cells, mitochondria were elongated tubules and were organized largely as parallel arrays that extend along the thin filaments (Fig. 5A) (Schultz et al, 2017). In the P316L variant muscle cells, the average mitochondrial area was increased (Fig. 5A,B). In addition, the width of the minor tubule axis was also increased (Fig. 5B), giving a more rounded or globular morphology (Fig. 5A,B). No difference in the long (major tubule) axis was noted between P316L variant and P316P control (Appendix Fig. S4D), suggesting that mitochondrial fragmentation did not account for this difference. The mitochondria of the deletion heterozygotes were not different from the control heterozygotes (Fig. 5A,B). These results indicate that the mitochondrial morphology is altered in the P316L variant, and likely also in the R167Q and S331F variants.

## Proband variants, p.Arg182Gln, p.Ser346Phe and p.Leu109Ser, disrupt ATP synthase (Complex V) enzyme activity

Next, we evaluated mitochondrial function in skin fibroblasts from probands 1, 4, and 6 representing the variants p.Arg182Gln, p.Ser346Phe, and p.Leu109Ser, respectively (Fig. 6A). Fibroblasts from the other probands were not available for study. On blue native PAGE, complex V migrated as a single band in all three probands as in the controls. Compared to 50 control fibroblast lines, the in-gel activity of complex V was markedly reduced in proband 6 (p.Leu109Ser), moderately reduced in proband 1 (p.Arg182Gln) and very mildly reduced in proband 4 (p.Ser346Phe) (Fig. 6B). In-gel activities of other complexes (I, II, and IV) were normal. This pattern in *ATP5F1A* fibroblasts is different than that noted for *ATP5PO* or in *MT-ATP6* variants where in the setting of Coomassie blue stained gels, the separation of the peripheral stalk results in complex V bands of lower MW (Ganapathi et al, 2022; Smet et al, 2009; Wittig et al, 2010), whereas for *ATP5F1A* probands, similar to *ATP5F1D* probands, the catalytic activity is reduced at the intact complex V without lower MW bands (Ganapathi et al, 2022; Olahova et al, 2018).

The complex V ATPase hydrolytic enzyme activity in fibroblasts was decreased in all three variant cell lines at 42% (p.Arg182Gln),

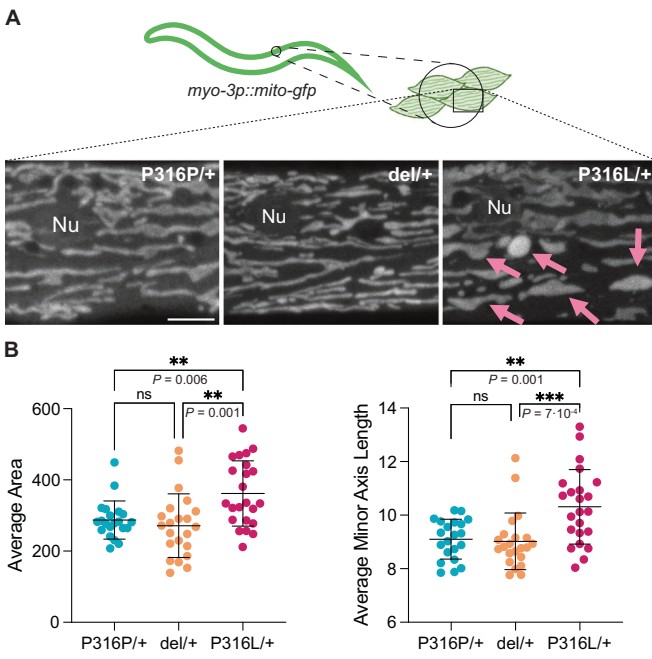

**Figure 5.** *atp-1* **P316L variant alters mitochondrial morphology.**

Mitochondrial morphology is altered in P316L animals. (A) Representative images of mitochondrial marker *bcIs80* (*myo-3p::mito-gfp*) expressed in the body wall muscles. Arrows point to abnormal mitochondria. Nu marks the position of the nucleus. Scale bar, 5 μm. (B) Quantification of average area and minor axis length per animal, calculated using MitoSegNet. Data are from 3 biological replicates. Each data point represents measurements from a single animal. The exact number of animals analyzed for each genotype is shown in Appendix Table S5. Mean and SD are shown. One-way ANOVA performed followed by post-hoc Holm–Sidak tests. ns not significant, **$P < 0.01$, ***$P < 0.001$. Source data are available online for this figure.

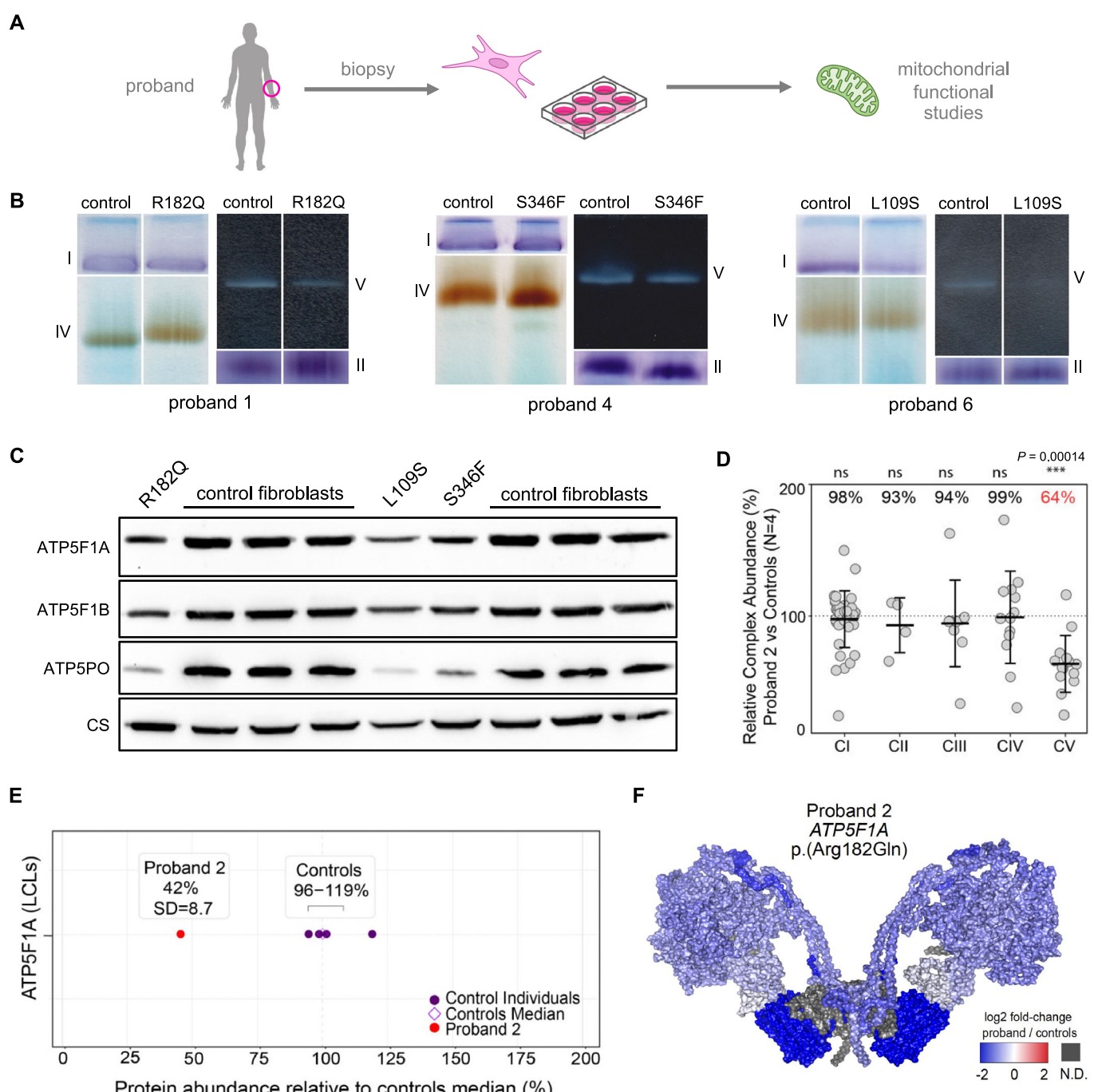

**Figure 6.  Proband fibroblasts show reduced complex V abundance and activity.**

(A) Schematic of mitochondrial functional studies performed on proband fibroblasts. (B) BN-PAGE analysis of complexes of the electron transport chain of proband 1 (p.Arg182Gln), proband 4 (p.Ser346Phe) and proband 6 (p.Leu109Ser) fibroblasts. (C) Western blots analysis of ATP5F1A, ATP5F1B and ATP5PO of proband and control fibroblasts. cs, citrate synthase, loading control. (D) Relative Complex Abundance (RCA) of OXPHOS complexes from proteomics data of Proband 2 (p.Arg182Gln) LCLs compared to controls ($N = 4$) showing an isolated Complex V defect. Middle bar represents mean complex abundance. Upper and lower bars represent 95% confidence interval. Significance was calculated from a two-sided $t$ test between the individual protein means. CI: $P = 0.23136$; CII: $P = 0.49824$, CIII: $P = 0.40071$, CIV: $P = 0.50014$, CV: $P = 0.00014$. (E) Protein range for ATP5F1A in LCLs of Proband 2 (p.Arg182Gln) (red dot) and controls ($N = 4$, purple dots) showing a standard deviation of 8.7 from the control median. (F) Topographical heatmap of the log2 fold-change abundances of Proband 2 relative to controls onto the cryo-EM structure of the dimer complex V structure (PDB: 7AJD). Source data are available online for this figure.

**Table 2. Respiratory chain complex abundances in fibroblasts of patients with complex V deficiency and controls.**

| Gene | Proband | Relative complex abundance | | | | | RCA ratios | | | Relative protein levels | | |
|---|---|---|---|---|---|---|---|---|---|---|---|---|
| | | CI | CII | CIII | CIV | CV | CI/CII | CIV/CII | CV/CII | ATP5F1A | ATP5F1D | ATP5PO |
| ATP5F1A | 1 | 112 | 106 | 128 | 95 | 58 | 1.057 | 0.896 | 0.547 | 48%; −7.5 | 58%; −2.5 | 39%; −7.2 |
| ATP5F1A | 4 | 110 | 97 | 132 | 115 | 53 | 1.134 | 1.186 | 0.546 | 44%; −8.4 | 51%; −3.1 | 34%; −8.6 |
| ATP5F1A | 6 | 115 | 102 | 138 | 117 | 58 | 1.127 | 1.147 | 0.569 | 44%; −8.4 | 69%; −1.6 | 34%; −8.4 |
| ATP5F1D | | 101 | 101 | 110 | 105 | 32 | 1.000 | 1.040 | 0.317 | 19%; −18.0 | 32%; −5.7 | 16%; −15.3 |
| ATP5PO | | 101 | 104 | 124 | 103 | 56 | 0.971 | 0.990 | 0.538 | 95%; +0.3 | 103%; +0.6 | 13%; −16.9 |
| Controls N = 23 | | | | | | | | | | | | |
| Mean ± SD | | 97.4 ± 7.3 | 97.1 ± 9.4 | 100.3 ± 6.3 | 99.6 ± 15.1 | 96.5 ± 7.5 | 1.01 ± 0.12 | 1.03 ± 0.20 | 1.03 ± 0.20 | | | |
| Range | | 87–114 | 77–110 | 88–114 | 83–150 | 84–114 | 0.84–1.36 | 0.83–1.79 | 0.81–1.36 | 76–110% | 65–97% | 69–116% |

*CI complex I, CII complex II, CIII complex III, CIV complex IV, CV complex V.*
*The relative complex abundances of the respiratory chain complexes and their internal ratios, and the relative amounts of the individual proteins as determined by proteomics analysis of fibroblasts after correction for mitochondrial abundance are listed as percent of in-assay controls (n = 5) for our probands. These are shown in comparison to a range of 23 normal controls and to cell lines of patients with other nuclear encoded autosomal recessive complex V defects. The mean, the standard deviation (SD) and the observed range among controls are provided. The relative protein levels are shown as percent of in assay controls; and as the Z-score of the distribution observed in normal control population. This control data did not deviate significantly from a normal distribution. Patient results significantly outside the range observed in control subjects are shown in bold. C/CII is the ratio of the RCA of complex I divided by the RCA of complex II. The RCAs of the large subunit of the mitoribosome, the small subunit of the mitoribosome and the pyruvate dehydrogenase complex were all normal and are therefore not shown.*

33% (p.Ser346Phe), and 37% (p.Leu109Ser) of the average of control cells (Appendix Table S2). These enzyme activities were similar to those previously reported for the dominant p.Arg207His variant (Lines et al, 2021; Zech et al, 2022), but more than those reported for probands with the recessive p.Arg329Cys variant (29% and 19%, R. Rodenburg, personal communication) (Jonckheere et al, 2013).

Western blot analysis of different complex V subunits indicated that ATP5F1A (α-subunit) and ATP5F1B (the β-subunit that associates with ATP5F1A) were moderately reduced while ATP5PO (the OSCP-subunit that also associates with ATP5F1A and ATP5F1B), was strongly reduced in all three probands (Fig. 6C).

Mass spectrometry based proteomic analysis of proband 2 (p.Arg182Gln) lymphoblastoid cells (LCLs) revealed a reduction of almost all complex V subunits (64% control mean) as illustrated in relative complex abundance (RCA) analysis (Figs. 6D and EV2A). Consistent with the western blot data, the abundance of the ATP5F1A protein in proband 2 was reduced to 42% of in assay controls median (Fig. 6E). On blue native PAGE, complex V migrated as a single markedly reduced band in proband 2's LCLs (Fig. EV2B), similar to proband 1 fibroblasts carrying the same p.Arg182Gln variant (Fig. 6B). Likewise, proteomic analysis of fibroblasts of probands 1, 4 and 6 revealed a comparable decrease in the RCA of complex V in all three probands (53–58%), and in the RCA of complex V over complex II (Table 2). There was a mild increase in the RCA of complex III. Similar to western blot findings, the ATP5F1A protein was reduced to 44–48% of controls whereas the ATP5PO protein in proband 1, 4, and 6 was reduced to 34–39% of controls (Table 2).

Since *ATP5F1A* is expected to be tolerant to loss-of-function variants (pLI=0.29; o/e = 0.3–0.58), a ~50% reduction in abundance of ATP5F1A is not predicted to be damaging. Therefore, the marked reduction in complex V enzyme activity (below 50% that of controls) and the observed reduction in abundance of both ATP5F1A-interacting F1 and distal Fo complex V subunits (as reflected in Figs. 6F and EV2C) indicates a destabilizing effect of the proband variants on complex V assembly and function, consistent with a dominant negative mechanism. Interestingly, dominant negative *ATP5F1A* proband fibroblasts or LCLs showed a milder reduction in complex V subunits when compared to fibroblasts of probands with biallelic loss of function variants in *ATP5F1D* (Olahova et al, 2018) and *ATP5PO* (Ganapathi et al, 2022).

## Decreased ATP and mitochondrial membrane potential despite increased oxygen consumption suggests uncoupled oxidative phosphorylation

To better understand the pathogenic mechanism of the *ATP5F1A* variants, we measured oxygen consumption, mitochondrial membrane potential, and steady state ATP in cells. On the Seahorse 96XF bioanalyzer (Agilent) mitochondrial stress test comparison of the proband 1 p.Arg182Gln variant fibroblasts and age-, sex-, and race-matched control fibroblasts, the proband 1 fibroblasts displayed significantly higher rates of basal respiration (i.e., unstressed) and an overall higher capacity for maximal respiratory function (Fig. 7A,B; Appendix Fig. S5). As a result, the spare respiratory capacity in proband fibroblasts was also overall

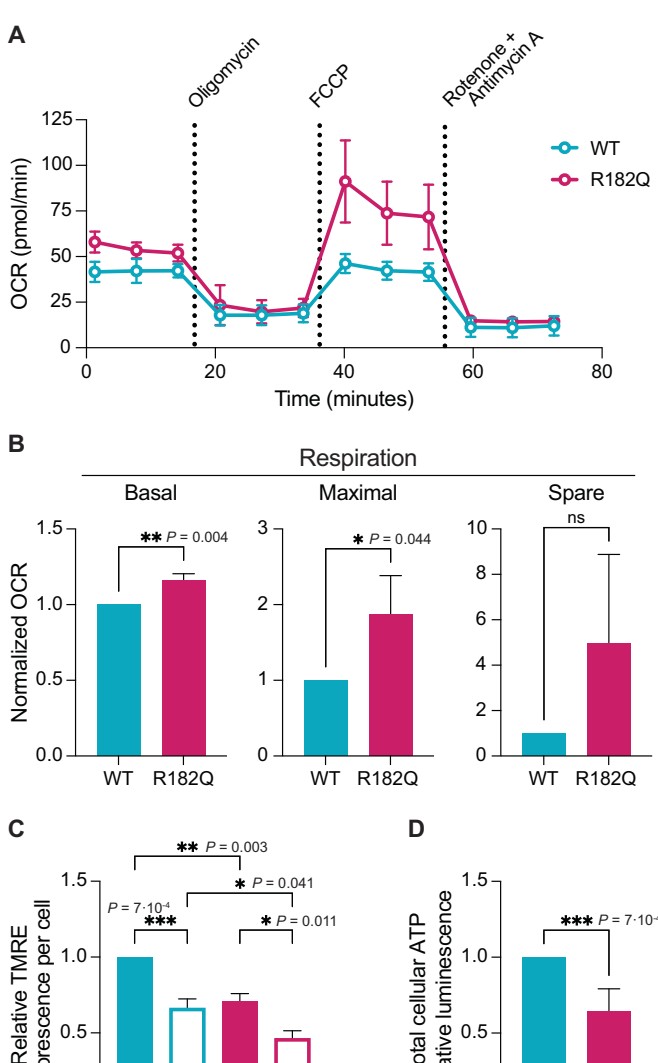

**Figure 7.** *ATP5F1A* p.R182Q proband fibroblasts show reduced ATPase activity and uncoupled oxidative phosphorylation.

(A) Representative oxygen consumption rate (OCR) of proband-derived fibroblasts compared to age, race, and sex matched control fibroblasts measured by Seahorse in intact cells. Mean and SD of one representative experiment is shown. Individual graphs from each of the three biological replicates are shown in Appendix Fig. S5. (B) Normalized average basal, maximal, and spare OCR from three biological replicates. Mean and SEM of three biological replicates are shown. Significance was calculated using an unpaired t-test. Individual graphs from each experiment are shown in Appendix Fig. S5. (C) Membrane potential of proband and control fibroblasts as measured by TMRE fluorescence with and without FCCP treatment. Mean and SEM of five biological replicates are shown. Significance was calculated from an ordinary One-way ANOVA analysis. (D) Total cellular ATP content taken from lysed proband and control fibroblasts using a luciferase reporter. Mean and SEM are of five biological replicates are shown. Significance was calculated using an unpaired t test. ns not significant, *$P < 0.05$, **$P < 0.01$, ***$P < 0.001$. Source data are available online for this figure.

higher, albeit not significant, compared to wild-type fibroblasts (Fig. 7B; Appendix Fig. S5). Despite the increased rates of oxygen consumption, the membrane potential of proband fibroblast mitochondria measured by the fluorogenic dye, tetramethylrhodamine ethyl ester (TMRE), was significantly lower than that of wild-type control (Fig. 7C) and was as low as that of wild-type fibroblasts treated with the uncoupling agent carbonyl cyanide-p-trifluoromethoxyphenylhydrazone (FCCP) (Fig. 7C). Furthermore, the proband fibroblasts exhibited significantly diminished ATP steady state levels compared to wild-type controls (Fig. 7D). These results are opposite to those reported for the p.Arg207His variant where the cells exhibited a lower rate of oxygen consumption and no change in mitochondrial membrane potential compared to control cells (Lines et al, 2021).

Collectively, these results indicate that in fibroblasts the inefficient ability of the *ATP5F1A* p.Arg182Gln variant to generate ATP is mostly the result of uncoupled oxidative phosphorylation. Cells use the normal activities of complexes I through IV resulting in higher oxygen consumption, but their defective complex V results in loss of membrane potential and decreased ATP.

## Discussion

We describe a series of 6 new probands with 4 heterozygous de novo missense variants in *ATP5F1A*, who presented with intellectual disability, developmental delay, prominent motor symptoms with variable degrees of dystonia and spasticity. Structural analysis suggested that the four variants likely disrupt the ATP5F1A's (α-subunit) interactions with the β− (ATP5F1B) and/or γ− (ATP5F1C) subunits of ATP synthase. Functional studies in *C. elegans* determined that all variants evaluated were damaging and act by a dominant negative mechanism. Analysis of proband-derived cells showed reduced ATPase activity and decreased abundance of ATP5F1A and other complex V subunits. For two proband variants, uncoupled oxidative phosphorylation was likely a mechanism of disease. Our study brings the total described families with dominant de novo missense variants in *ATP5F1A* currently to 12, making this one of the most common causes of nuclear genome encoded complex V deficiency together with recessive *TMEM70* variants (Honzik et al, 2010; Magner et al, 2015). Furthermore, except for *ATP5F1A* and *ATP5F1B*, autosomal recessive mode of inheritance is currently the only known genetics for complex V deficiency associated with *ATPAF2*, *ATP5F1E*, *ATP5PO*, *ATP5MD*, *ATP5F1D* and *TMEM70*. Thus, the autosomal dominant mode of *ATP5F1A* pathogenicity, and likely also for *ATP5F1B*, may have been previously underappreciated. From a clinical diagnostic perspective, this study also illustrates that if a variant of uncertain significance is identified in *ATP5F1A*, multiple methods can potentially confirm the pathogenicity including: (1) blue native PAGE with in-gel activity staining, (2) hydrolytic ATPase complex V enzyme assay, (3) proteomics studies to identify both the reduced abundance of complex V via RCA and of the ATP5F1A protein, (4) mitochondrial respiration (Seahorse) analysis, and, 4) functional studies in model organisms. It is worth noting that the reduction in abundance of ATP5F1A and other complex V proteins observed in this study is similar to what is

observed in variants in other parts of the catalytic central globular head of complex V, such as in *ATP5F1D*, and that identification of the correct molecular cause remains paramount for a final diagnosis (Fig. EV2C). On the other hand, the impact of dominant negative *ATP5F1A* variants on complex V subunit abundance is clearly different from that of recessive variants in *ATP5F1D* and *ATP5PO* (Fig. EV2C), where the central stalk remains preserved.

Although so far rare in mitochondrial complexes, a dominant negative effect can be observed in proteins that form hetero-multimers and can involve disrupting interactions between subunits (Anders and Botstein, 2001; Bejsovec and Anderson, 1988; Kemphues et al, 1980; Lund et al, 1997). The catalytic head domain of ATP synthase consists of 3 α- and 3 β-subunits, which interact with the central stalk (γ-subunit). Protein modeling of ATP synthase showed that all four residues, Arg182, Ser346, Pro331, and Leu109, affected in the probands are located at or near the contact points of α:β and/or α:γ subunits of the $F_1$ (Figs. 2 and EV1). Therefore, substitution of these residues with the proband variant is predicted to alter the protein conformation and disrupt the interactions between the α−, β− and/or γ−subunits leading to altered ATP synthase function. Indeed, our BN-PAGE, western blot, electron-transport chain enzyme assays, and proteomic analysis showed reduced activity and abundance of complex V subunits. Since *ATP5F1A* is not a haploinsufficient gene (pLI = 0.29, gnomAD v4.1), the observed complex V defects cannot be explained simply by ~50% reduction in function or abundance of ATP5F1A. We postulate a dominant negative mechanism whereby the proband variants negatively affect complex V and alter its function and stability. Our hypothesis is based on the supposition that 1/8th, 3/8th, 3/8th and 1/8th of the complex V consists of 3, 2, 1 and 0 variant α-subunits, respectively (see Appendix Fig. S6). We propose that complexes having 2 or 3 variant α-subunits (4/8th or 50%) are likely unstable and undergo degradation while 3/8th of the complex comprising of only 1 variant α-subunit is stable but has reduced function. ATP5PO which interacts with both the α-subunit and the β−subunit, was strongly reduced with impact on the stability of the side chain. The remaining 1/8th of the complex comprising of all wild-type α-subunits functions normally. Thus, the variable levels of ATP5F1A and complex V observed in our probands likely reflects the variable activity, stability and dominant negative effect of the variant protein on the conformation of the assembled complex V. A similar dominant negative effect was recently postulated for a variant in *ATP5F1B*, which encodes the β-subunit of ATP synthase (Fig. EV1) (Ganetzky et al, 2022; Nasca et al, 2023). However, in contrast to *ATP5F1B* probands (Ganetzky et al, 2022), the *ATP5F1A* probands described here showed no clinical evidence of hypermetabolism. Of note, the dominant negative de novo *ATP5F1A* cases have less severe clinical presentation than the homozygous cases (Jonckheere et al, 2013; Lieber et al, 2013). This is likely because in the dominant negatives cases, 1/8th of complex V which comprises of all wild-type α-subunits is fully functional.

Dominant de novo missense variants in *ATP5F1A* have previously been reported, however, functional evaluation has only been described for one variant, p.Arg207His (Lines et al, 2021; Zech et al, 2022). Four individuals with this variant presented with life-threatening, neonatal onset mitochondriopathy with failure to thrive, hyperlactatemia and hyperammonemia. Interestingly, all four showed spontaneous improvement and clinical resolution in late infancy with no persistent neurological phenotypes. Biochemical analysis of p.Arg207His fibroblasts showed reduced complex V ATPase activity (<20–30% of controls), reduced ATP5F1A subunit protein abundance (Lines et al, 2021; Zech et al, 2022), but lower oxygen consumption rate with no change in mitochondrial membrane potential (Lines et al, 2021). All probands in our study also showed a similar reduction in ATPase activity (~33–42% of controls) and reduced abundance of ATP5F1A and other complex V subunits. However, in contrast to p.Arg207His proband, our p.Arg182Gln proband (proband 1) fibroblasts showed higher basal and maximal respiration rates and reduced mitochondrial membrane potential suggesting uncoupled oxidative phosphorylation. Although the exact molecular mechanism remains to be determined, the uncoupled oxidative phosphorylation is similar to that reported for the dominant negative variant in *ATP5F1B* (Ganetzky et al, 2022), thus representing a shared mechanism for dysfunction in the catalytic head, that is distinct from that of *MT-ATP* variants. Moreover, the distinct clinical presentations and mitochondrial physiology findings between p.Arg207His individuals and our p.Arg182Gln proband suggests that dominant variants in *ATP5F1A* can cause disease via additional distinct genetic and molecular mechanisms.

Additional support for uncoupled oxidative phosphorylation in our probands comes from our *C. elegans* studies that showed altered mitochondrial morphology. The elongated tubular mitochondria of the wild-type body wall muscle cells became more rounded and globular in the *atp-1*[P316L] variant animals. Interestingly, cultured cells treated with compounds that uncouple the mitochondria or cause loss of mitochondrial membrane potential displayed similar changes in mitochondrial morphology from a tubular to a globular shape (Miyazono et al, 2018). Although the changes observed in cultured cells may not be directly comparable to those in *C. elegans* muscle cells, these results lend further support for mitochondrial uncoupling in *atp-1*[P316L] mitochondria. Of note, *atp-1*[P316L] corresponds to *ATP5F1A* p.Pro331Leu (proband 5) and *atp1* P328L in yeast, a known mitochondrial genome integrity (*mgi*) mutant (Chen and Clark-Walker, 1995, 1996; Clark-Walker et al, 2000; Wang et al, 2007). All *mgi* mutants identified so far are without exception, dominant, map to α, β, or γ subunits of ATP synthase, and uncouple oxidative phosphorylation (Wang et al, 2007). These observations suggest that oxidative phosphorylation is also uncoupled in our p.Pro331-Leu (proband 5) variant and indicate that uncoupled oxidative phosphorylation may be a common feature in our probands. Analogous findings were recently reported by Ganetzky et al, who described a dominant negative, uncoupling, missense *ATP5F1B* variant in monozygotic twins with a higher oxygen consumption rate and lower mitochondrial membrane potential (Ganetzky et al, 2022). Interestingly, the *ATP5F1B* variant is located near a known *mgi* locus, and the twins have failure to thrive and developmental delay, similar to the probands in this study. Therefore, yeast conserved legacy *mgi* mutants could provide helpful functional data in determining the pathogenicity and genetic mechanism of ATP synthase variants that may in the future be identified in human patients.

In summary, using an integrated multidisciplinary approach which included structural modeling, genetic and cell biology studies in *C. elegans*, and biochemical and mitochondrial physiology studies in patient cells, we provide strong evidence that de novo

heterozygous variants, p.Arg182Gln, p.Ser346Phe, p.Pro331Leu, and p.Leu109Ser, in *ATP5F1A* are damaging and function via a dominant negative mechanism. Comparison of our cases with others reported in the literature suggested that not all dominant de novo *ATP5F1A* variants work via the same genetic and/or molecular mechanism, underscoring the importance of in vivo and in vitro functional analyses for a correct molecular diagnosis. There may also be differences in pathophysiological mechanism between cell types, as pertinent differences between fibroblasts and neuronal cells have been reported in some complex V deficient patients (Lorenz et al, 2017). We recommend that a similar integrated approach be used for characterizing future novel variants in *ATP5F1A* and other ATP synthase subunits to obtain a more complete understanding of the variant mechanism, which is a prerequisite for developing an effective treatment strategy. There are now 12 individuals reported with de novo heterozygous variants in *ATP5F1A*, making *ATP5F1A* the most frequent nuclear genome cause of complex V (ATP synthase) deficiency. Our study expands the spectrum of *ATP5F1A*-associated conditions to include dominant negative variants that likely uncouple oxidative phosphorylation and ATP production.

# Methods

### Reagents and tools table

| Reagent/resource | Reference or source | Identifier or catalog number |
| --- | --- | --- |
| **Experimental models** | | |
| *C. elegans* strains | This study, Appendix Table S3 | |
| Control Primary fibroblast | Coriell Biobank | GM03349 |
| **Recombinant DNA** | | |
| pRF4 plasmid | Mello et al (1991) | |
| **Antibodies** | | |
| ATP5F1A (ATP5A) | Abcam | ab110273 |
| ATP5F1B (ATP5B) | Abcam | ab128743 |
| ATP5PO (ATP5O) | Abcam | ab110276 |
| Citrate synthase | Abcam | ab129095 |
| ATP5F1A | Abcam | ab14748 |
| SDHA | Abcam | ab14715 |
| **Oligonucleotides and other sequence-based reagents** | | |
| CRISPR gRNAs, repair templates | This study, Appendix Table S1 | |
| **Chemicals, enzymes and other reagents** | | |
| Aminocaproic acid | Fisher Scientific | 103301000 |
| Serva G | Helixx Technologies | 35050 |
| Nitro blue tetrazolium | Sigma | N6878 |
| NADH | Fisher Scientific | 27110010 |
| 3,3'-diaminobenzidine | Sigma | C9322 |
| Cytochrome c | Sigma | C7752 |
| Catalase | Sigma | D8000 |
| Pb(NO3)2 | Fisher Scientific | L62 |
| ATP | Sigma | A2383 |
| Cas9 protein | IDT | 1081060 |
| **Software** | | |

| Reagent/resource | Reference or source | Identifier or catalog number |
| --- | --- | --- |
| Wormlab | MBF Bioscience; https://www.mbfbioscience.com/products/wormlab/ | |
| MitoSegNet | https://github.com/MitoSegNet/MitoS-segmentation-tool | |
| Fiji (ImageJ) | https://fiji.sc/ | |
| Grid/Collection Stitching plugin in Fiji | https://imagej.net/plugins/grid-collection-stitching | |
| Prism | GraphPad | |
| Spectronaut (19.6.250122.62635, Sagan) | Biognosys; https://biognosys.com | |
| R (4.3.0) | https://www.r-project.org | |
| R Studio (2023.03.1 + 446) | https://www.r-studio.com | |
| Perseus (2.0.10.0) | https://maxquant.net/perseus/ | |
| PyMol (2.1.1) | https://www.pymol.org | |
| **Other** | | |
| OrbiTrap Mass Spectrometer (Astral/Eclipse) | Thermo Fischer Scientific | |
| Seahorse XF Cell Mito Stress Test Kit | Agilent | 103015-100 |
| Seahorse XFe96/XF Pro FluxPak | Agilent | 103792-100 |
| Seahorse XF RPMI assay medium pack, pH 7.4 | Agilent | 103681-100 |
| Seahorse XF 96-well Bioanalyzer | Agilent | |
| S-Trap™ Micro columns | ProtiFi, LLC | C02-micro |
| Luminescent ATP Detection Assay Kit | Abcam | ab113849 |
| TMRE-Mitochondrial membrane Potential Assay Kit | Abcam | ab113852 |

# Consent for genetic study

Informed consent was obtained from all human subjects, and the experiments conform to the principles set out in the WMA Declaration of Helsinki and the Department of Health and Human Services Belmont Report. Families of probands 1 and 5 provided informed consent for participation in the Undiagnosed Diseases Network (UDN), under a protocol approved by the NIH IRB (15HG0130). Proband 2 was enrolled in the Rare Disease Now Program at the Murdoch Children's Research Institute (HREC 67401) (Australia) and parents provided informed consent for genome sequencing, proteomic studies and publication of clinical and molecular data. Written informed consent was obtained from participants and/or their legal guardians for proband 3 for the collection, research use, and storage of the specimens according to the protocol approved on July 1, 2020, by Indiana University Probands 1 and 4 were studied in Colorado under COMIRB protocol # 16-0146 after obtaining informed consent. Clinical deidentified data of Proband 6 was analyzed with IRB approval under Baylor College of Medicine study protocol (H-48670), and the fibroblast studies were ordered as part of his clinical care. Fibroblast studies in Colorado on Probands 1, 4 and 6 were further studied under a protocol approved by the Colorado Multiple Institutional Review Board COMIRB protocol # 18-1828.

## Sequencing and bioinformatic analysis

Trio exome or whole genome sequencing was performed for probands 1, 5 and 6 by Baylor Genetics as described previously (Keehan et al, 2021). Variant filtering was performed using Codified Genomics software. Variants with an allele frequency of >0.01 in gnomAD (v.2.1.1) were filtered out. De novo, compound heterozygous, and X-linked single nucleotide and small indel variants were prioritized for review. Trio WGS for proband 2 was processed by the Centre for Population Genomics (Australia) using the DRAGEN GATK best practices pipeline. Reads were aligned to the hg38 reference genome with Dragmap (v1.3.0), and joint variant calling across the cohort was performed using GATK HaplotypeCaller (v4.2.6.1) for SNVs and indels. Variants were annotated with VEP 110 and integrated into Seqr (Pais et al, 2022) for filtration and analysis. For patients 3 and 4, initial clinical exome sequencing was performed by GeneDX (Gaithersburg, MD), identifying a heterozygous variant of uncertain significance (VUS) in ATP5F1A [(NM_004046.6:c.1037 C > T, p.(Ser346Phe)]. For patient 3, subsequent parental testing confirmed that this variant arose de novo. To further investigate the genetic findings, research-based whole genome sequencing (WGS) data re-analysis was conducted as described (Jacobs et al, 2022) using the EMEDGENE analytical pipeline (https://iuhealth.emedgene.com/) and supplemented with manual curation. Identified variants were filtered based on minor allele frequency (<0.1% in gnomAD), conservation, and deleteriousness by in silico prediction tools (PolyPhen-2, MutationTaster, REVEL, DANN, LRT, and SIFT).

## 3D molecular modeling

To explore the impact of the missense variants identified in this study we employed 3D molecular modeling as described (Booth et al, 2020; Booth et al, 2025). Briefly, previously described structural models comprising the 10-subunit human ATP synthase (PDB: 8H9S), were obtained (Lai et al, 2023). Alignment, visualization, and mutagenesis were performed by using PyMOL (version 2.5.5). Note single letter amino acid abbreviations were used to denote amino acids of interest.

## *C. elegans* strains and culturing conditions

Single letter amino acid abbreviations were used to denote *atp-1* variants studied in *C. elegans*. P316L and S331F variant lines and corresponding controls (P316P and S331S) were maintained and all assays for these lines were performed at 20 °C on NGM plates, while R167Q variant and R167R control lines were maintained and all assays except the development assay (Fig. 3D) were performed at 15 °C as the strong embryonic and larval lethality of this variant was partially suppressed at this temperature. Strains used in this study were as follows: *tmC5* (balancer for the *atp-1* extra copy on IV), *hIn1* (balancer for *atp-1* deletion and variants), *bcIs80* [*myo-3p::mito-gfp* + pRF4] (a generous gift from Barbara Conradt, unpublished), *zcIs13* [*hsp-6::gfp*], and *atp-1(ok2203)*. A full list of strains their genotypes are listed in Appendix Table S3.

## Transgene generation

Using Recombination Mediated Cassette Exchange (RMCE), an extra wild-type copy of *atp-1* was inserted as a single copy onto chromosome

IV (Nonet, 2020). This extra copy had a modified PAM sequence used for the R167 variant and control edits so that Cas9 would only edit the endogenous copy of *atp-1 and* had an additional silent NciI restriction site added in order to differentiate the extra copy on IV from the endogenous copy on I in RNA-seq (Appendix Table S4). To generate the P316 and S331 variants, the extra copy was first edited using the corresponding control edit repair template to destroy the PAM for each locus, and subsequently the newly generated variant edits and control edits at the endogenous locus were crossed back to the original extra copy line.

Since all variant edited strains were homozygous lethal, they were maintained over a balancer chromosome (see Appendix Fig. S7). Due to the infertility of the heterozygotes, all variant strains were also maintained with one extra copy of *atp-1*, over a second balancer chromosome. The extra copy and balancer for the endogenous copy were removed by crossing prior to phenotyping (Appendix Fig. S7).

## RNAseq and analysis

To determine the relative expression level from the *atp-1* extra copy transgene compared to the endogenous *atp-1* gene, RNA-seq was performed with strain udnSi40 (Appendix Table S3), which is homozygous for both *atp-1* loci, followed by read analysis as previously described (Marom et al, 2023). Two single-nucleotide silent polymorphisms in the *atp-1* extra copy transgene were used to distinguish transcripts from the two *atp-1* loci. While the *atp-1* transgenic extra copy was found to be expressed at about 50% the level of the endogenous *atp-1* gene (Appendix Fig. S3), the wild-type extra copy gene can still be used in gene dosage studies to assess whether one or two additional wild-type copies can suppress the dominant, change of function, *atp-1* variant alleles.

## CRISPR-Cas9 gene editing

CRISPR-Cas9 editing was performed as previously described (Fielder et al, 2022; Huang et al, 2022). Before knocking in the P316L and S331F variants, we first introduced two silent base pair changes to the transgenic *atp-1* extra copy to facilitate CRISPR-Cas9 editing of the endogenous *atp-1* locus and to prevent editing of the transgenic copy. Briefly, edited *atp-1*[extra copy] animals 24 h post larval L4 stage were injected with Cas9 protein (IDT 1081060), tracrRNA (IDT 1072533), pRF4 plasmid as a positive injection control, and IDT Alt-R® CRISPR-Cas9 sgRNA (Appendix Table S1A) and repair template (Appendix Table S1B). Repair templates were 100 bp in length, centered around the proband variant. Silent changes were made to eliminate the PAM, and to add or eliminate a restriction site (FokI eliminated in R167, XbaI added in S331, and HaeIII added in P316). To ensure any phenotypes observed were due to the proband variants and not the silent changes, control repair templates were used that only contained silent changes (referred to in text as P316P, S331S, and R167R). Variant edit lines were crossed back to the original extra copy line so that all lines were maintained with the same extra copy.

## Development assays

L4 double balanced heterozygotes with one extra copy were mated with wild-type males for 24 h at 20 °C (Appendix Fig. S7). Next,

mated animals were allowed to lay embryos on an empty plate for 2 h and then removed. 40 Embryos for controls or 90 embryos from variants were moved to a separate plate and allowed to grow at 20 °C for 72 h. Twenty-four hours after embryo laying, remaining embryos on plates were scored as dead embryos. At 72 h post embryo laying, stages of animals and genotypes were quantified. Four replicates were performed. For R167Q, 35–70 embryos were used as mated animals laid less than 90 total embryos.

## Length, crawling, and thrashing assays

Wormlab experiments were performed as previously described (Huang et al, 2022). Briefly, double heterozygous animals were mated to VC2010 males and progeny allowed to grow at 20 °C or 15 °C (R167 group) (Appendix Fig. S7). All animals assayed were heterozygous for *tmC5* balancer, aside from VC2010 control (Appendix Fig. S7). 1-day-old adult progeny were gently transferred to an NGM plate with a thin lawn of OP50 bacteria. Animals were allowed to adjust to recover from transferring for at least 30 min, and then were recorded for 2 min on the plate. After all crawling plates had been recorded, one milliliter of PBS was added to plates, and animals were recorded swimming for 2 min. Videos were processed by WormLab (MBF Bioscience) to detect and track both crawling and swimming animals. Animals recorded crawling continuously for 30 s or swimming for 20 s continuously were kept for analysis.

## Mitochondrial stress assays in *C. elegans*

The transgene *Phsp-6::gfp* was used to measure mitochondrial stress (Yoneda et al, 2004). Double balanced heterozygous animals with one extra copy were crossed with wild-type (VC2010), or *atp-1*(extra copy) animals to cross out the *hIn1* balancer and to only have one copy of the *tmC5* balancer except for animals with two extra copies (Appendix Fig. S7). In total, 8–10 animals 24 h post L4 stage were placed per well in a 398-well plate with PBS + 1% pluronic acid that had been stored at the experimental temperature (20 °C for P316 and S331, or 15 °C for R167). Overall, 2.5 mg/mL Levamisole in PBS stored at the experimental temperature was added to each well for a final concentration of 1.5 mg/mL at the same time and immediately taken for imaging in the brightfield and GFP fluorescence channel on the Thermofisher CellInsight CX-7 HCS as previously described (Gosai et al, 2010; Li et al, 2014; Long et al, 2014). The same images were run through two separate image thresholds, one where GFP head fluorescence was the only fluorescence counted in *tmC5/+* animals (labeled as WT in graphs), and a second where GFP head fluorescence + fluorescence in the intestine of control animals was detected. Average head fluorescence per worm was determined from WT animals. Average head fluorescence for one WT animal was multiplied by number of animals in each well and subtracted from total fluorescence to obtain only the stress fluorescence. Total stress fluorescence was normalized by area of animals to correct for differences in animal size, and then normalized to WT fluorescence to control for differences between replicates.

## Mitochondrial morphology in *C. elegans* muscle

Larval stage L4 animals homozygous for *bcIs80* and *tmC5* balancer, and heterozygous for the *hIn1* balancer were imaged with a ×63 oil objective lens at ×3 magnification on 2% agarose pads in 10 mM levamisole. Muscles from the dorsal two quadrants were imaged approximately halfway between the pharynx and the vulva. Animals were imaged within 15 min of being placed in levamisole. Maximum projections of 4–6 µm z-stacks were used. Z-projections were smoothed once in ImageJ and then processed by the ImageJ MitoSegNet PreProcessing plug in to set image depth. Images were then processed by MitoS CPU for windows program with a 20 pixel minimum size to create image masks, and then analyzed for mitochondrial size using MitoA analyzer and compared using Prism (Fischer et al, 2020). In total, 20–23 images were used for each genotype.

## Mitochondrial assays in fibroblasts

Respiratory chain enzyme activities for complexes I, II, III, II + III, IV and citrate synthase were assayed spectrophotometrically at 30 °C in cellular homogenates as previously described and compared to over 30 control fibroblasts (Chatfield et al, 2015; Coughlin et al, 2015). Complex V activity was determined by measuring the oligomycin-sensitive ATP synthase hydrolytic activity as detailed previously (Mayr et al, 2004; Rustin et al, 1994). In this assay, ATP is converted to ADP and $P_i$ by ATP synthase that is coupled via pyruvate kinase and lactate dehydrogenase to the oxidation of NADH to $NAD^+$. The oxidation of NADH was followed at 340 nm for 5 min at 37 °C, after which oligomycin was added and the assay was followed for an additional 5 min to determine the oligomycin-insensitive ATPase activity.

Solubilized inner mitochondrial membrane fractions were isolated by differential centrifugation from skin fibroblasts and were separated by blue native polyacrylamide gel electrophoresis (BN-PAGE) and then evaluated by in-gel activity staining assays for complexes I, II, IV, and V as described previously (Chatfield et al, 2015; Coughlin et al, 2015; Smet et al, 2009; Van Coster et al, 2001). The assembly of complex I by non-denaturing BN-PAGE followed by western blot analysis using the antibody against NDUFS2 (1:2000 dilution) was also analyzed as described (Friederich et al, 2017).

Steady state amounts of the proteins of subunits of complex V catalytic core and peripheral stalk including ATP5PO, the linker between the $F_1$ catalytic core and the peripheral stalk, ATP5F1A and ATP5F1B, the α and the β subunits of the $F_1$ catalytic core, respectively, were assessed by SDS-PAGE followed by western blot as previously described (Ganapathi et al, 2022). Dilutions of antibodies used for western blots were as follows: anti-ATP5PO (1:2000), anti-ATP5F1A (1:2000), anti-ATP5F1B (1:1000) and anti-citrate synthase (1:2000).

The maximal catalytic ATP synthesis rate was measured based on the method described with some modifications (Rustin et al, 1994). Fibroblasts were permeabilized with digitonin (2 µg/mL for 1 min), and incubated with 1 mM substrates malate and pyruvate, and 0.2 mM DDAP. The reaction was started by addition of excess ADP 1 mM at 37 °C with sampling at time points 0, 1 min, 2 min, 3 min, 5 min, and 15 min. Samples were then deproteinized with PCA, pH neutralized, and ATP was measured following derivatization to ethenoadenosine using bromoacetaldehyde (Sharon et al, 2006), and a Adenine nucleotides were separated using reverse phase HPLC while monitoring fluorescence at $\lambda_{exc}$ 280 nm $\lambda_{em}$ 410 nm. Bromoacetaldehyde was made by refluxing of bromoacetaldehyde diethylacetal in acetic acid environment at 100 °C for 2 h followed by neutralization to pH 4.3. Adenosine

containing analytes were derivatized to ethenoadenosine by combining with 2× volume of the bromoacetaldehyde solution (~245 mM) followed by incubation at 80 °C for 20 min, then cooled and filtered through 0.22 μm centrifugal membranes before HPLC on an Agilent 1100 series. The HPLC separation method involved: 10 μL injection sample volume, 0.7 mL/min flow rate; Atlantis® T3 3.0 ×100 mm column with 3 μM bead size (Waters) controlled at 30 °C. The gradient involved solvent A: 30 mM potassium phosphate pH 5.45 with 0.8 mM tetrabutylammonium phosphate (TBAP, an ion pairing reagent); and solvent B: 1:1 volume by volume of acetonitrile: 30 mM potassium phosphate pH 7 with 0.8 mM TBAP in water. After optimization, the following gradient was used: 90/10% A/B from 0 to 0.1 min, change to 70/30% from 0.1 to 0.7 min, hold at 70/30% from 0.7 to 1.7 min, change to 55/45% from 1.7 to 2.6 min, hold at 55/45% from 2.6 to 5 min, before returning back from 5 to 6 min to initial 90/10% and re-equilibrating for 20 min. Analytes were detected using an Agilent 1200 series fluorescence detector with excitation λ 280 nm and emission λ 410 nm. All adenosine species was resolved with retention times: ADO ~3.2 min; AMP ~4.1 min; ADP ~5.8 min; ATP ~6.7 min, and two alternative internal standards: 2-methyl-adenosine ~3.5 min, and 5-acyl-adenosine ~4.0 min. Using 5 repeat measurements of a standard curve run on 3 separate days, the limit of detection was 500 nM and the limit of quantification was 1 μM for all analytes, and the coefficient of variation for repeat analyses was <1%. The appearance rate of ATP production was used with subtraction of the oligomycin-resistant rate.

## Blue native PAGE in lymphoblastoid cell lines

Blue Native PAGE (BN-PAGE) analysis was performed by solubilizing whole cell lymphoblastoid pellets from the proband 2 and two unrelated individual pediatric controls in a digitonin solution containing solubilization buffer (20 mM Bis-Tris (pH 7.0), 50 mM NaCl, 10% glycerol, 1% digitonin (w/v)). Protein concentration was determined using the Pierce Protein Assay Kit (Thermo Fisher Scientific), and a total of 30 μg of whole cell lysate was separated on an Invitrogen NativePAGE Bis-Tris Gel (3–12%), followed by transferring proteins onto PVDF membrane (Merck) using the Invitrogen Mini Blot Module transfer system, as per the manufacturer's instructions. The gel loading was visualized by staining the PVDF membrane in a stain containing 0.1% Coomassie Brilliant Blue G-250 (CBB; Acros Organics). Immunoblots were developed using primary antibodies for ATP5F1A (Abcam; ab14748) and SDHA (Abcam; ab14715) at 1:1000 dilutions and secondary horseradish peroxidase coupled mouse antibody (Cell Signalling Technology) at 1:10,000 dilution. Images were visualized on a ChemiDoc XRS+ imaging machine (BioRad) using Clarity Western ECL Substrate (BioRad).

## Seahorse analysis of proband fibroblasts

The MitoStress test was conducted on a Seahorse 96XF bioanalyzer according to the manufacturer's instruction. After performing optimization assays on wild-type fibroblasts for cell number (3000, 10,000, 15,000 and 20,000 cells) and trifluoromethoxyphenylhydrazone (FCCP) concentration (0.25, 0.5 and 1 μM), 15,000 cells per well (seeded 16–24 h prior to running on the bioanalyzer) and 1 μM FCCP were used for all MitoStress Tests comparing

fibroblasts from *ATP5F1A* p.R182Q proband and race, sex and age-matched control fibroblasts (GM03349, Coriell Institute for Medical Research).

## TMRE-mitochondrial membrane potential measurements of proband fibroblasts

*ATP5F1A* p.R182Q proband or control fibroblast cells were cultured in DMEM medium supplemented with 15% heat-inactivated FBS (complete medium). Cells were seeded at a density of $1 \times 10^4$ cells/well of a 96-well plate. Following an overnight incubation, mitochondrial membrane potential was measured using the TMRE-Mitochondrial membrane potential assay kit (Abcam AB11385) following the manufacturer's protocol. Briefly, culture medium was removed and replaced with either complete medium or 20 μM FCCP for 10 min. TMRE at a final concentration of 20 nM was added to cells and incubated for 30 min. Medium was then replaced with 100 μl of PBS/0.2% BSA + 5 μg/ml Hoechst and incubated for 20 min before imaging. TMRE fluorescence (peak emission 575 nM) was detected by the CX-7 high-content imager (Thermofisher). The cell number was determined by counting the number of nuclei using the Hoechst mask. TMRE fluorescence was normalized to the cell number. All data were further normalized to the untreated wild-type control.

## Cellular ATP measurements of proband fibroblasts

Total cellular ATP was measured using the Luminescent ATP Detection Assay Kit (Abcam, AB113849) according to the manufacturer's instructions. Briefly, *ATP5F1A* p.R182Q proband or control fibroblast cells were cultured in 96-well plates at a density of $1.4 \times 10^4$ cells/well. The next day, detergent (provided in the kit) was added to lyse the cells and stabilized the ATP. Following a 5-min incubation, the substrate solution was added to each well. Plates were wrapped in aluminum foil and further incubated for 10 min before quantification. Luminescence was measured on the TECAN Infinite M Plex.

## Acquisition and analysis of mass spectrometry data

Proteomics analysis in fibroblasts was performed using the patient sample in triplicate and five in-assay controls as described in detail previously (Van Hove et al, 2024). Briefly, after washing fibroblasts in PBS after harvesting, were resuspended in SDS-TEAB buffer, and protein measured (Pierce BCA Protein Assay Kit). Twenty-five microgram protein was reduced and alkylated using tris(2-carboxyethyl)phosphine (TCEP) and 2-chloroacetamide, transferred to an S-Trap micro column (Protifi, LLC), and overnight trypsinized. The peptides are eluted from the S-Trap micro column using three elution buffers, dried and resuspended in 2% acetonitrile 0.1% trifluoroacetic acid and 3 μL per sample was loaded on an OrbiTrap Eclipse Mass Spectrometer (Thermo Fisher Scientific) for liquid chromatography–tandem mass spectrometry (LC-MS/MS) analysis. After loading onto an Acclaim Pepmap 100 C18 75 μm × 2 cm trapping column, they were chromatographically resolved on-line using an EASY-Spray Pepmap RSLC C18 75 μm × 25 cm 2 μm analytical column (Thermo Scientific) using an Ultimate 3000 RSLCnano LC system. Data were acquired in data-independent acquisition (DIA) experiments, full MS resolutions were set to 120,000 at *m/z* with data acquired in centroid mode using positive polarity in 50 windows with resolution set to 30,000.

Raw data were processed using the Spectronaut platform (version 19.6.250122.62635, Sagan, Biognosis) and searched against the UniProt human database (canonical peptides + isoforms, reviewed, 42,447 entries), excluding single hit proteins. For proband 2, lymphoblastoid cells from proband (in triplicate) and four unrelated controls were prepared for proteomics as previously described (Hock et al, 2024). LC-DIA-MS was carried out using an Orbitrap Astral mass spectrometer (Thermo Scientific) equipped with a Vanquish Neo UHPLC (Thermo Scientific) and using the heated trap and elute setup. The columns were Acclaim Pepmap nano-trap column (Dionex—C18, 100 Å, 75 μm × 2 cm) and 5.5 cm high-throughput uPAC Neo analytical column (Thermo Scientific). The eluents were water with 0.1% v/v formic acid (solvent A) and 80% CH3CN with 0.1% v/v formic acid (solvent B). The peptides were separated on the analytical column using the following gradient at a flow rate of 750 nl/min—(i) 0–0.3 min 3-6% B, (ii) 0.3–23 min, 6–23.5% B (iii) 23–26.7 min, 23.5–40% B (iv) 26.7–28.7 min 40-50% B (v) 28.7–28.8 min, 50–99% B and (vi) 28.8–0 min 99%. Column was then equilibrated for 7 min before the next sample injection. For DIA experiments, full MS resolutions were set to 120,000 and scanning from 380 to 980 $m/z$ in the profile mode. Full MS AGC target was 500% with a maximum IT of 5 ms. DIA was carried out in the Astral analyzer with isolation window of 2 $m/z$, normalized HCD collision energy of 27, normalized AGC target of 500% and maximum injection time of 3 ms. The cycle time was kept to 0.6 s.

MS2 Quantity from protein groups were imported into Perseus (version 2.0.10.0) and two valid values were filtered for patient and in-assay controls. $\text{Log}_2$ transformed data was generated in Perseus using MitoCarta3.0 (Rath et al, 2021) + entries and normalization via subtract row cluster means prior to a two-sample $t$ test using $P$ value = 0.05 for truncation. Log2 fold-changes were applied to the topographical heatmap using a custom build script (Stroud et al, 2016) against the bovine complex V structure downloaded from the protein data bank PDB: 7AJD (Berman et al, 2000; Spikes et al, 2021). The relative complex abundance (RCA) was calculated and plotted for each complex of the respiratory chain, the small and the large subunit of the mitoribosome and the pyruvate dehydrogenase complex in R (version 4.3.0) and RStudio (version 2023.03.1 + 446) using a custom build program (Hock et al, 2024). The ATP5F1A range plot was calculated in R using MS2 quantities from ATP5F1A normalized to the mean of mitochondrial proteins annotated with MitoCarta3.0 and relative to the median value of controls as previously described (Hock et al, 2024). To determine the variation in normal controls, the RCA was developed from 23 deidentified normal control fibroblast cells with verified normal standard cellular energetics assays.

## Statistics

For *C. elegans* data (Figs. 3, 4, and 5), statistical analyses were performed in GraphPad Prism (version 10). Data sets were first tested for normality, then an ANOVA (for normal data), or Kruskal-Wallis (for data sets with at least one non-normal genotype) followed by post-hoc Holm–Sidak testing were performed between genotypes. For patient cell data (Figs. 6 and 7), statistical significance was determined using two sample $t$ test as outlined in the legend. Studies performed in this manuscript were unblinded and randomization procedures were not used. Data are from a minimum of three biological replicates are reported unless otherwise noted in the figure legend. The number of biological replicates and total $n$ for each experiment are shown in Appendix Table S5.

## The paper explained

### Problem

Advances in whole genome sequencing technology and bioinformatics tools have accelerated the identification of candidate disease gene variants. However, determining the causality of rare genetic variants remains challenging.

### Results

Rare dominant genetic variants in the gene, *ATP5F1A*, were identified in six individuals who exhibited overlapping symptoms including developmental delay, intellectual disability, and movement disorders. To understand the consequences of these genetic variants we performed functional studies in the nematode *C. elegans*. Animals carrying the patient variants displayed abnormal development, slow locomotion, and atypical mitochondrial morphology indicating that these variants disrupted gene function. Moreover, these phenotypes were suppressed by increasing the dose of the wild-type gene, indicating that this group of variants disrupted function through a dominant negative mechanism. Studies in patient-derived cells revealed that the variants disrupted mitochondrial respiratory chain function by reducing complex V activity and uncoupling oxidative phosphorylation. Collectively, the results indicated that the gene variants are damaging and likely responsible for the patients' symptoms.

### Impact

This study provides strong support for the pathogenicity of the rare dominant variants identified in the six individuals and emphasizes the importance of functional studies in understanding the molecular, genetic, and cellular mechanisms of disease.

## Data availability

This study includes data deposited in external repositories. Mass spectrometry proteomics data for patients 1, 2 and 4 were deposited into Zenodo (www.zenodo.org) under the https://doi.org/10.5281/zenodo.16124448. The data from patient 6 was obtained under a deidentified protocol that does not allow for deposition into a public repository.

The source data of this paper are collected in the following database record: biostudies:S-SCDT-10_1038-S44321-025-00290-8.

## Peer review information

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

## Acknowledgements

The authors would like to thank the subjects and their families for participation in the study. The authors would also like to thank Mike Nonet for helpful consultation in using RMCE to make the atp-1[extra copy] line, and the Genome Technology Access Center (GTAC), Washington University in St. Louis for assistance with the RNA-seq studies. Some strains were provided by the CGC, which is funded by NIH Office of Research Infrastructure Programs (P40 OD010440). Research reported in this manuscript was also supported by the NIH Common Fund, through the Office of Strategic Coordination/Office of the NIH Director under award number U54 NS108251 (TS and LSK), from the National Human Genome Research Institute U01HG007709 and U01HG007942. This project was supported in part by the Center for Rare, Undiagnosed and Genetic Diseases (St Louis Children's Hospital Foundation) (GAS and SCP), NIH NIGMS R35 GM152192 (TS), the Clinical Translational Core (CTC) of the Baylor College of Medicine (BCM) Intellectual and Developmental Disabilities Research Center (IDDRC). The BCM IDDRC is supported by P50 HD103555 from the Eunice Kennedy Shriver National Institute of Child Health and Human Development (NICHD). This study was also funded in part by the Indiana University Grand Challenge Precision Health Initiative. In Colorado, the study was supported by philanthropic support from the Children's Hospital Colorado Summits for Samantha, and the University of Colorado Foundation, and by a grant from the National Institutes of Health, NIH U54NS078059 for the North American Mitochondrial Disease Consortium (NAMDC) to JLKVH NAMDC is part of Rare Diseases Clinical Research Network (RDCRN), an initiative of the Office of Rare Diseases Research (ORDR), NCATS. This consortium is funded through collaboration with NCATS. The work performed in the lab of RR is supported by NIH/NCCR 1 S10 OD028538-01A1 to Nichole Reisdorph. In Melbourne, the study was supported by an Australian National Health and Medical Research Council (NHMRC) Investigator Fellowship (2009732 to DAS) along with funding by Medical Research Future Fund Genomics Health Futures Mission (2016030 to DAS and DHH). We thank the Mito Foundation for the provision of instrumentation through research equipment grants to DAS and DHH and the Melbourne Mass Spectrometry and Proteomics Facility (MMSPF) for the provision of instrumentation and training. Analysis was supported by the Centre for Population Genomics (Garvan Institute of Medical Research and Murdoch Children's Research Institute) and was funded in part by a National Health and Medical Research Council investigator grant (2009982). The Rare Disease Flagship acknowledges financial support from the Royal Children's Hospital Foundation [2019-1198 and 2023-1484], the Murdoch Children's Research Institute, Paula Fox, The Andrew and Geraldine Buxton Foundation and The Pierce Armstrong Foundation. The research conducted at the Murdoch Children's Research Institute was supported by the Victorian Government's Operational Infrastructure Support Program.

## Author contributions

**Sara M Fielder**: Data curation; Formal analysis; Validation; Investigation; Methodology; Writing—original draft; Writing—review and editing. **Marisa W Friederich**: Data curation; Investigation; Visualization; Methodology; Writing—review and editing. **Daniella H Hock**: Data curation; Formal analysis; Investigation; Visualization; Methodology; Writing—review and editing. **Jessie R Zhang**: Formal analysis; Investigation; Writing—review and editing. **Liana M Valin**: Investigation; Writing—review and editing. **Jill A Rosenfeld**: Resources; Investigation; Writing—review and editing. **Kevin T A Booth**: Investigation; Visualization; Writing—review and editing. **Natasha J Brown**: Resources; Investigation; Writing—review and editing. **Rocio Rius**: Investigation; Writing—review and editing. **Tanavi Sharma**: Investigation; Writing—review and editing. **Liana N Semcesen**: Investigation; Writing—review and editing. **Kim C Worley**: Investigation; Writing—review and editing. **Lindsay C Burrage**: Investigation; Writing—review and editing. **Kayla Treat**: Resources; Investigation; Writing—review and editing. **Tara Samson**: Investigation; Writing—review and editing. **Sarah Govert**: Investigation; Writing—review and editing. **Sara DaCunha**: Investigation; Writing—review and editing. **Weimin Yuan**: Investigation; Writing—review and editing. **Jian Chen**: Investigation; Writing—review and editing. **Jacob Lesinski**: Investigation; Writing—review and editing. **Hieu Hoang**: Investigation; Writing—review and editing. **Stephanie A Morrison**: Visualization; Writing—review and editing. **Farah A Ladha**: Resources; Investigation; Writing—review and editing. **Roxanne A Van Hove**: Investigation; Writing—review and editing. **Cole R Michel**: Investigation; Writing—review and editing. **Richard Reisdorph**: Investigation; Writing—review and editing. **Eric Tycksen**: Data curation; Writing—review and editing. **Dustin Baldridge**: Data curation; Writing—review and editing. **Gary A Silverman**: Supervision; Funding acquisition; Writing—review and editing. **Claudia Soler-Alfonso**: Resources; Supervision; Writing—review and editing. **Erin Conboy**: Resources; Supervision; Writing—review and editing. **Francesco Vetrini**: Resources; Supervision; Writing—review and editing. **Lisa Emrick**: Resources; Supervision; Writing—review and editing. **William J Craigen**: Resources; Supervision; Writing—review and editing. **Stephen M Sykes**: Formal analysis; Supervision; Validation; Project administration; Writing—review and editing. **David A Stroud**: Supervision; Funding acquisition; Validation; Project administration; Writing—review and editing. **Johan L K Van Hove**: Conceptualization; Supervision; Funding acquisition; Validation; Writing—original draft; Project administration; Writing—review and editing. **Tim Schedl**: Conceptualization; Supervision; Funding acquisition; Validation; Writing—original draft; Project administration; Writing—review and editing. **Stephen C Pak**: Conceptualization; Supervision; Funding acquisition; Validation; Writing—original draft; Project administration; Writing—review and editing.

Source data underlying figure panels in this paper may have individual authorship assigned. Where available, figure panel/source data authorship is listed in the following database record: biostudies:S-SCDT-10_1038-S44321-025-00290-8.

## Disclosure and competing interests statement

The authors declare no competing interests.

## Undiagnosed Diseases Network

Jill A Rosenfeld[8], Kim C Worley [iD][8], Lindsay C Burrage[8,13], Kayla Treat[9], Dustin Baldridge[1], Erin Conboy[9], Francesco Vetrini[9], Lisa Emrick[7,13], William J Craigen[7,13], Tim Schedl [iD][14,17] and Stephen C Pak [iD][1,17 ✉]

A full list of members and their affiliations appears in the Supplementary Information.

# Expanded View Figures

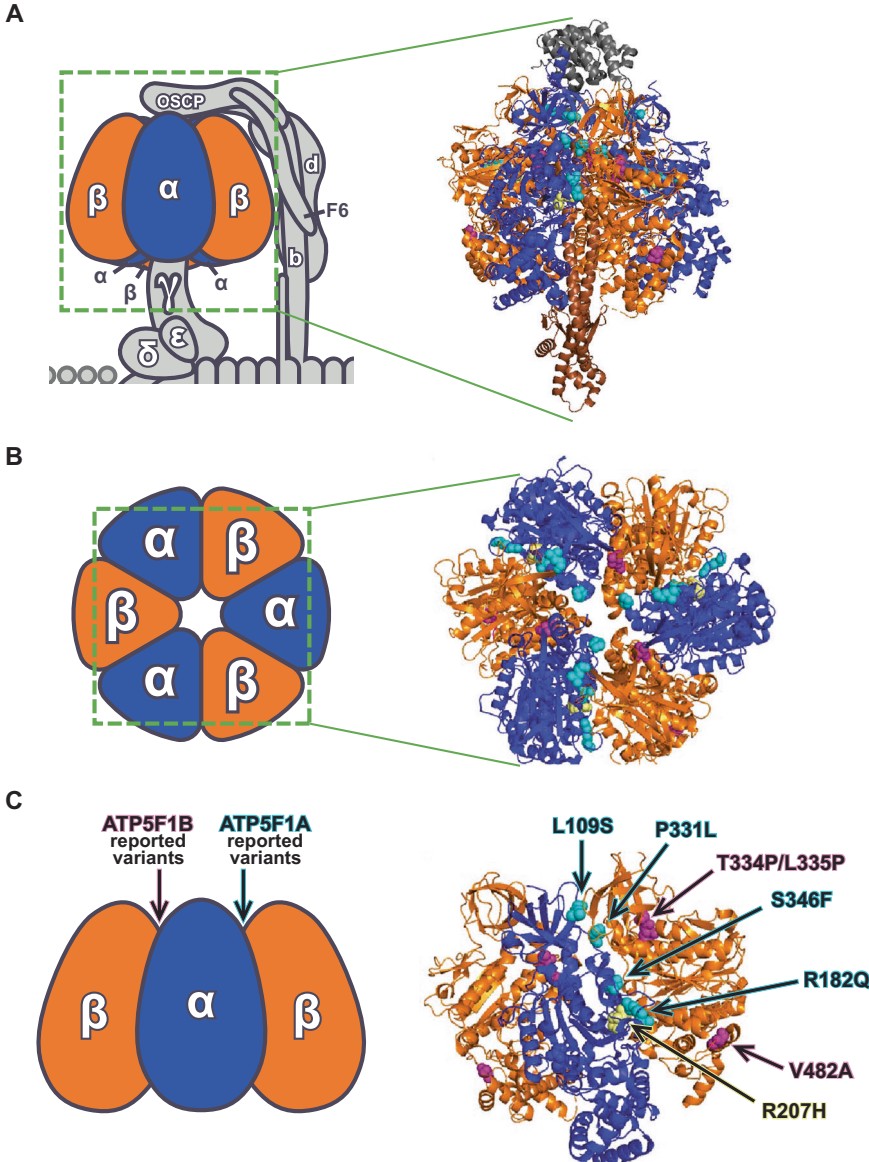

**Figure EV1. 3D molecular modeling of dominant variants in ATP5F1A and ATP5F1B.**

(A) Side and (B) top-down views of the cryo-structure of α- and β-subunits of ATP synthase. (C) A zoomed in view of the β:α:β subunits. The *ATP5F1A* variants (L109S, R182Q, P331 and S346F) from this study are shown in cyan, the previously published *ATP5F1A* variant (R207H) and *ATP5F1B* variants (T334P, L335P, V482A) are shown in yellow and magenta, respectively. Alignment, visualization, and mutagenesis were performed by using PyMOL (version 2.5.5) using previously described structural models comprising the 10-subunit human ATP synthase (PDB: 8H9S) (Lai et al, 2023).

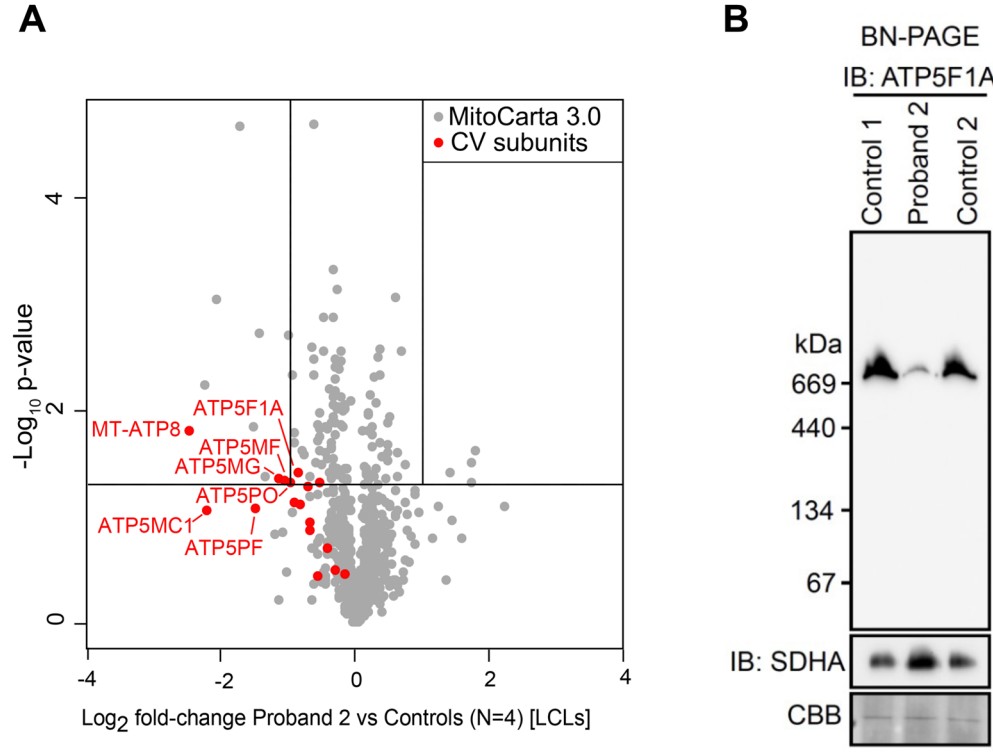

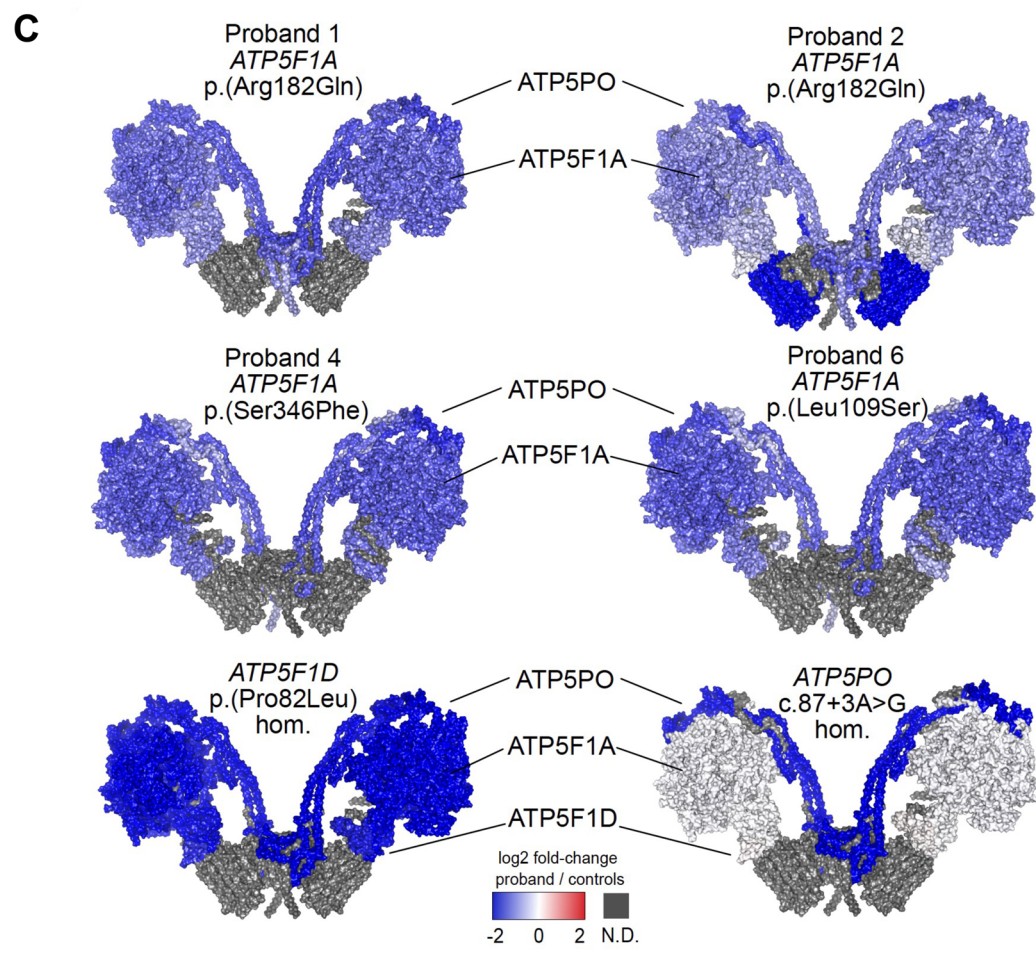

◄ **Figure EV2. Proteomics analysis of mitochondrial proteins.**

(A) Volcano plot of mitochondrial proteins annotated from MitoCarta3.0 of Proband 2 (p.R182Q) lymphoblastoid cell lines (LCLs) compared to controls ($N = 4$) showing reduced abundance of subunits of Complex V. Vertical lines represent ± 2-fold-change equivalent and horizontal lines represent significance $P$ value = 0.05 equivalent from a two-sample $t$ test. Red = Complex V subunits. (B) Blue native PAGE and immunoblotting (IB) of LCLs from Proband 2 and two unrelated controls against ATP5F1A and SDHA antibodies showing reduced abundance of complex V in Proband 2. CBB: Coomassie Brilliant Blue. (C) Topographical heatmap of the log2 fold-change abundances onto the cryo-EM structure of the dimer complex V structure for probands 1, 2, 4 and 6 as well as disease controls with known biallelic pathogenic variants in *ATP5F1D* and *ATP5PO* relative to controls. The topographical heatmaps are coloured using the fold-changes from the t-test results obtained in (A). Source data are available online for this figure.

