## [Peer Review File · EMBO Molecular Medicine]

Dominant negative ATP5F1A variants disrupt oxidative phosphorylation causing neurological disorders

Sara Fielder, Marisa Friederich, Daniella Hock, Jessie Zhang, Liana Valin, Jill Rosenfeld, Kevin Booth, Natasha Brown, Rocio Rius, Tanavi Sharma, Liana Semcesen, Kim Worley, Lindsay Burrage, Kayla Treat, Tara Samson, Sarah Govert, Sara DaCunha, Weimin Yuan, Jian Chen, Jacob Lesinski, Hieu Hoang, Stephanie Morrison, Farah Ladha, Roxanne Van Hove, Cole Michel, Richard Reisdorph, Eric Tycksen, Dustin Baldrige, Gary Silverman, Claudia Soler-Alfonso, Erin Conboy, Francesco Vetrini, Lisa Emrick, William Craigen, Undiagnosed Diseases Network, Stephen Sykes, David Stroud, Johan Van Hove, Tim Schedl, and Stephen Pak

Corresponding author: Stephen Pak (stephen.pak@wustl.edu)

Review Timeline:

Submission Date:	26th Mar 25
Editorial Decision:	28th Apr 25
Revision Received:	8th Jul 25
Editorial Decision:	18th Jul 25
Revision Received:	24th Jul 25
Accepted:	25th Jul 25

Editor: Zeljko Durdevic

Transaction Report:

28th Apr 2025

Dear Dr. Pak,

Thank you for the submission of your manuscript to EMBO Molecular Medicine. We have now received feedback from the two reviewers who agreed to evaluate your manuscript. Both referees recognize interest of the study but also raise important concerns that should be addressed in a major revision. If you would like to discuss further the points raised by the referees, I am available to do so via email or video. Let me know if you are interested in this option.

We would welcome the submission of a revised version within three months for further consideration. Please let us know if you require longer to complete the revision.

I look forward to receiving your revised manuscript.

Yours sincerely,

Zeljko Durdevic

We require:

- 1) A .docx formatted version of the manuscript text (including legends for main figures, EV figures and tables). Please make sure that the changes are highlighted to be clearly visible.
- 2) Individual production quality figure files as .eps, .tif, .jpg (one file per figure). For guidance, download the 'Figure Guide PDF': (<https://www.embopress.org/page/journal/17574684/authorguide#figureformat>).
- 3) A .docx formatted letter INCLUDING the reviewers' reports and your detailed point-by-point responses to their comments. As part of the EMBO Press transparent editorial process, the point-by-point response is part of the Review Process File (RPF), which will be published alongside your paper.
- 4) A complete author checklist, which you can download from our author guidelines (<https://www.embopress.org/page/journal/17574684/authorguide#submissionofrevisions>). Please insert information in the checklist that is also reflected in the manuscript. The completed author checklist will also be part of the RPF.
- 5) Please note that all corresponding authors are required to supply an ORCID ID for their name upon submission of a revised manuscript.
- 6) It is mandatory to include a 'Data Availability' section after the Materials and Methods. Before submitting your revision, primary datasets produced in this study need to be deposited in an appropriate public database, and the accession numbers and

database listed under 'Data Availability'. Please remember to provide a reviewer password if the datasets are not yet public (see <https://www.embopress.org/page/journal/17574684/authorguide#dataavailability>).

12) Author contributions: You will be asked to provide CRediT (Contributor Role Taxonomy) terms in the submission system. These replace a narrative author contribution section in the manuscript.

13) A Conflict of Interest statement should be provided in the main text.

14) Every published paper now includes a 'Synopsis' to further enhance discoverability. Synopses are displayed on the journal webpage and are freely accessible to all readers. They include a short stand first (maximum of 300 characters, including space) as well as 2-5 one-sentences bullet points that summarizes the paper. Please write the bullet points to summarize the key NEW findings. They should be designed to be complementary to the abstract - i.e. not repeat the same text. We encourage inclusion of key acronyms and quantitative information (maximum of 30 words / bullet point). Please use the passive voice. Please attach these in a separate file or send them by email, we will incorporate them accordingly.

15) Include a Reagents and Tools Table as part of the Methods section, which can be downloaded from our author guidelines (<https://www.embopress.org/page/journal/17574684/authorguide#structuredmethods>)

***** Reviewer's comments *****

Referee #1 (Comments on Novelty/Model System for Author):

The authors have extensively tested their hypothesis in a *C. elegans* model and in cultured patient fibroblasts. The choice of model organism is convincing.

Referee #1 (Remarks for Author):

Fielder et al present a compelling study in which they describe a novel mechanism by which heterozygous de novo dominant mutations in the ATP5F1A gene cause a mitochondrial disease in six patients carrying four different ATP5F1A variants. Most of the probands manifested with dystonia and/or spasticity and developmental delay.

This is a very carefully executed study that uses structural argument and a *Caenorhabditis elegans* animal model to demonstrate that mutations at the interacting surfaces / contact points of the alpha-subunit with neighboring structures of the F1F0-ATPase cause dysfunction and destabilization of complex V thereby reducing ATP synthesis, increasing uncoupling and reducing the mitochondrial membrane potential in patient fibroblasts. This stands in stark contrast to previously described recessive disease mutations in the same gene.

The authors show in the worm model that the dominant negative effect can be overcome by increasing the number of wildtype *atp-1* mRNA copies in the cells. They also show that haploinsufficiency by removing one entire allele of the *atp-1* gene (homolog of ATP5F1A) did not show any phenotype. Further they investigated in their worm model the presence of mitochondrial stress by visualizing the unfolded protein response with a *hsp-6p::gfp* reporter and the mitochondrial morphology.

I just have some minor suggestions:

[1] It would be of interest to know, whether the p.Leu335Pro (Ganetzky et al. 2022) and p.Thr334Pro, p.Val482Ala (Nasca et al. 2023) in ATP5F1B would also be located at contact point between alpha- und beta-subunits. The position of this mutation could be highlighted in Figure 2, giving more arguments in favor of the authors hypothesis on pathophysiology.

Please also locate the de novo mutations described by Zech et al. 2022 and Lines et al. 2021 on the cryo-structure of Figure 2.

[2] The fact that the here described dominant mutations are found at the contact points between the subunits should be mentioned in the abstract.

[3] As the majority of the here described patients presented with dystonia I would like to see their cranial MRI scans in the supplementary section of the manuscript, preferably the T2-weighted images of the basal ganglia/striatum and the brain stem.

Referee #2 (Comments on Novelty/Model System for Author):

This is an interesting well written paper on the discovery of several de novo heterozygous missense mutations in the gene encoding ATP5F1A subunit of mitochondrial *atp*-synthase. The alpha subunit encoded by ATP51A works in combination. with the beta subunit forming dynamically open-close structures moved by the central stalk of ATPsynthse during its rotation caused by the movement of the c-subunit cylinder actioned by the passage of protons through the oblique channel largely formed by the mtDNA subunit 6. The mutations found in this paper are in close proximity if not in junction with residues of the beta subunit. This suggests that the effects of the missense mutation in the alpha subunit can be expanded to malfunctioning of the beta subunit as well, therefore amplifying the effect of the mutation and explaining the dominant effect demonstrated also by different

and convincing approaches, including biochemistry, animal C. elegans) models, respirometry etc. The paper is multidisciplinary, experiment look impeccable, conclusions are convincing. I have a couple of questions: what was the main symptomatology prompting the Authors to suspect a mitochondrial disease? Was there a free window or the symptoms were congenital? Although this is not the case in the series presented here, other subjects with the mutation in the same gene showed improved age-related clinical conditions. Any hypothesis regarding this interesting phenomenon? Any hint that one of the patients of the present series may undergo the same clinical mitigation/improvement. Concerning the uncoupling phenomenon observed in the present series, shouldn't this determine any dysfunction in heat production, dysregulation of the body temperature, or any other phenomenon related to the discrepancy between increased oxygen consumption and lower than normal ATP production?

Referee #2 (Remarks for Author):

This is an impeccable piece of clinical and experimental work with several new findings and interesting conclusion about the damage of mitochondrial ATP synthase through a convincingly demonstrated dominant negative effect. I think the paper is worth to be published by EMBO Mol. Medicine.

Response to Referee's Comments

The authors would like to thank the reviewers for their thoughtful review of our manuscript. Below is a point-by-point response to each of the two reviewers' comments (in blue), with reference to specific revisions in the current version.

Referee #1

Fielder et al present a compelling study in which they describe a novel mechanism by which heterozygous de novo dominant mutations in the *ATP5F1A* gene cause a mitochondrial disease in six patients carrying four different *ATP5F1A* variants. Most of the probands manifested with dystonia and/or spasticity and developmental delay.

This is a very carefully executed study that uses structural argument and a *Caenorhabditis elegans* animal model to demonstrate that mutations at the interacting surfaces / contact points of the alpha-subunit with neighboring structures of the F1F0-ATPase cause dysfunction and destabilization of complex V thereby reducing ATP synthesis, increasing uncoupling and reducing the mitochondrial membrane potential in patient fibroblasts. This stands in stark contrast to previously described recessive disease mutations in the same gene.

The authors show in the worm model that the dominant negative effect can be overcome by increasing the number of wildtype *atp-1* mRNA copies in the cells. They also show that haploinsufficiency by removing one entire allele of the *atp-1* gene (homolog of *ATP5F1A*) did not show any phenotype. Further they investigated in their worm model the presence of mitochondrial stress by visualizing the unfolded protein response with a *hsp-6p::gfp* reporter and the mitochondrial morphology.

I just have some minor suggestions:

[1] It would be of interest to know, whether the p.Leu335Pro (Ganetzky et al. 2022) and p.Thr334Pro, p.Val482Ala (Nasca et al. 2023) in *ATP5F1B* would also be located at contact point between alpha- and beta-subunits. The position of this mutation could be highlighted in Figure 2, giving more arguments in favor of the authors hypothesis on pathophysiology. Please also locate the de novo mutations described by Zech et al. 2022 and Lines et al. 2021 on the cryo-structure of Figure 2.

As suggested by the reviewer, we have included the locations of the previously reported p.Arg207His variant in *ATP5F1A* (Lines et al. 2021 and Zech et al. 2022), and the p.Leu335Pro (Ganetzky et al. 2022) p.Thr334Pro, and p.Val482Ala (Nasca et al. 2023) variants in *ATP5F1B* on the cryo-structure. This information is presented in a new **Expanded View Figure (Fig. EV1)**. Six out of seven dominant variants reported so far are located at or near the contact points between *ATP5F1A* and *ATP5F1B* subunits highlighting the importance of protein conformation on coordinated interaction between the α - and β -subunits for ATP synthase function. Further

studies are required to better understand the structural and functional consequences of these variants and how they contribute to different clinical presentations.

[2] The fact that the here described dominant mutations are found at the contact points between the subunits should be mentioned in the abstract.

We have revised the **Abstract** to include that the variants are located at the contact points between the α - and β -subunits.

[3] As the majority of the here described patients presented with dystonia I would like to see their cranial MRI scans in the supplementary section of the manuscript, preferably the T2-weighted images of the basal ganglia/striatum and the brain stem.

MRIs for probands #1, #3, #5 and #6 were performed externally so images were not available. However, according to the clinical notes, MRIs of proband #1 and proband #3 were unremarkable (no comments on basal ganglia or brainstem). Proband #5, had stable right sided ventriculomegaly associated with posterior predominate white matter volume loss. Volume loss of the right hippocampus and suspected of the right mesial temporal lobe indicate concurrent right mesial temporal lobe sclerosis. Decreased perinasal sinus inflammatory changes now with opacification only in the right maxillary sinus. Proband #6 (latest MRI at age 5) was negative for acute intracranial, orbital or sinonasal abnormalities. Right-sided serous mastoid air cell effusion. MRIs for probands #2 and #4 were performed internally and are included in the revised manuscript as new **Appendix Fig. S1**. No abnormalities were noted.

Referee #2 (Comments on Novelty/Model System for Author):

This is an interesting well written paper on the discovery of several de novo heterozygous missense mutations in the gene encoding ATP5F1A subunit of mitochondrial atp-synthase. The alpha subunit encoded by ATP51A works in combination with the beta subunit forming dynamically open-close structures moved by the central stalk of ATPsynthse during its rotation caused by the movement of the c-subunit cylinder actioned by the passage of protons through the oblique channel largely formed by the mtDNA subunit 6. The mutations found in this paper are in close proximity if not in junction with residues of the beta subunit. This suggests that the effects of the missense mutation in the alpha subunit can be expanded to malfunctioning of the beta subunit as well, therefore amplifying the effect of the mutation and explaining the dominant effect demonstrated also by different and convincing approaches, including biochemistry, animal C. elegans) models, respirometry etc. The paper is multidisciplinary, experiment look impeccable, conclusions are convincing.

I have a couple of questions:

1) what was the main symptomatology prompting the Authors to suspect a mitochondrial disease?

Mitochondrial disease was not suspected in the probands as their presentations were relatively non-specific. Metabolic workup showed some elevated lactate and pyruvate, but overall profiles were not consistent with a known metabolic disorder. Mitochondrial disease was only suspected after the identification of candidate *ATP5F1A* variants following trio exome/whole genome sequencing and confirmed by functional studies.

2) Was there a free window or the symptoms were congenital?

For proband 4, the first noted symptom was developmental delays at age 1 year. For all other probands, symptom onset was congenital.

3) Although this is not the case in the series presented here, other subjects with the mutation in the same gene showed improved age-related clinical conditions. Any hypothesis regarding this interesting phenomenon?

This is certainly intriguing and not easily explained. Insufficient data are available to even generate a hypothesis. Prior to the report by Lines et al. 2021, transient symptoms in primary mitochondrial disorders were only noted for Reversible mitochondrial myopathy with COX deficiency and reversible hepatopathy in TRMU disorder. All other primary mitochondrial disorders are progressive. It will be interesting to hear from the authors of those cases what the long term follow up will be. There have been cases where symptoms have appeared many years after a transient period of apparent clinical resolution.

Any hint that one of the patients of the present series may undergo the same clinical mitigation/improvement.

Most of our probands are currently teenagers. Their conditions have either remained steady or worsened over time. None have shown signs of clinical improvement.

4) Concerning the uncoupling phenomenon observed in the present series, shouldn't this determine any dysfunction in heat production, dysregulation of the body temperature, or any other phenomenon related to the discrepancy between increased oxygen consumption and lower than normal ATP production?

Although hyperthermia was reported for the individuals with the uncoupling variant, *ATP5F1B* p.Leu335Pro (Ganetzky et al. 2022), dysregulation of body temperature was not reported for the probands in the current study. These observations suggest that uncoupling variants in core subunits of ATP synthase can present with different clinical phenotypes. Whether this is due to differences in the severity of the uncoupling or due to another mechanism remains to be determined. Of note, at the most recent United Mitochondrial Disease Foundation (UMDF) conference (June, 2025), other cases with some evidence of uncoupling were presented, none of them with hyperthermia.

Referee #2 (Remarks for Author):

This is an impeccable piece of clinical and experimental work with several new findings and interesting conclusion about the damage of mitochondrial ATP synthase through a convincingly demonstrated dominant negative effect. I think the paper is worth to be published by EMBO Mol. Medicine.

We thank the reviewer for their positive comments.

18th Jul 2025

Dear Dr. Pak,

Thank you for the submission of your revised manuscript to EMBO Molecular Medicine. I am pleased to inform you that we will be able to accept your manuscript pending the following final amendments:

- 1) Authors: Please define the consortium "Undiagnosed Diseases Network" using the attached guidelines. Please note, that the consortium should also be entered as an author in our submission system, if it is included in the main author list, together with the contact details of a nominated consortium representative.
- 2) Please address all comments suggested by our data editors listed below:
 - o Figure legends:
 1. Please note that the exact p values are not provided in the legends of figures 3F, G; 4B, C; 6D.
 2. Please indicate the statistical test used for data analysis in the legend of figure EV2 B.
 - Limit keywords to max. 5.
 - Add callouts for Figure EV1.
 - Add "Disclosure and competing interests statement". We updated our journal's competing interests policy in January 2022 and request authors to consider both actual and perceived competing interests. Please review the policy <https://www.embopress.org/competing-interests> and update your competing interests if necessary.
 - In Methods, provide the statement that informed consent was obtained from all human subjects and confirm that the experiments conformed to the principles set out in the WMA Declaration of Helsinki and the Department of Health and Human Services Belmont Report.
 - In Methods, provide antibody dilutions that were used for each antibody.
 - In Methods, a statistical paragraph should reflect all information that you have filled in the Authors Checklist, especially regarding randomization, blinding, replication.
 - Indicate in legends exact n and exact p values, not a range, along with the statistical test used. To keep the figures "clear" some authors found providing an Appendix table Sx with all exact p-values preferable. You are welcome to do this if you want to.
 - Raw data from large-scale datasets e.g. WES, RNA sequencing or Mass Spectrometry should be deposited in one of the relevant databases and made freely available prior the publication of the manuscript. The journal encourages authors to provide access to genotype and clinical data with as few restrictions as possible while respecting ethical obligations to the patients and relevant medical and legal issues. A signed statement of informed consent to publish any human clinical, large-scale, and genomic datasets must be obtained from each person (parents or legal guardians for minors) who appears in a study. Please check "Author Guidelines" for more information <https://www.embopress.org/page/journal/17574684/authorguide#datadeposition>
Use the following format to report the accession number of your data:

[data type]: [full name of the resource] [accession number/identifier] ([doi or URL or identifiers.org/DATABASE:ACCESSION])

Please check "Author Guidelines" for more information.

<https://www.embopress.org/page/journal/17574684/authorguide#availabilityofpublishedmaterial>

- Please correct the reference citation in the reference list. Where there are more than 10 authors on a paper, 10 will be listed, followed by "et al.". Also, please remove DOIs. DOIs should only be used for preprints and datasets that have not been published. Please check "Author Guidelines" for more information.

<https://www.embopress.org/page/journal/17574684/authorguide#referencesformat>

3) Figures: Please mark all splices in Figure 6B. Use a visible line or space (e.g., a thin white or black line) to indicate where the blot was spliced.

4) Reagent Table: Please correct callout "Table S3" to "Appendix Table S3".

5) Appendix: Please place the figure legends underneath the corresponding figure, add page numbers to the table of contents, and correct the nomenclature to "Appendix Table S1" etc.

6) Funding: Please ensure that all funding sources listed in the Acknowledgements are entered into our system - currently the National Institute of Neurological Disorders and Stroke of the National Institutes of Health U01HG007709 and U01HG007942, Children's Discovery Institute, St Louis Children's Hospital Foundation etc. are only mentioned in the manuscript text.

7) The Paper Explained: Please add it to the main manuscript file.

8) Synopsis:

- Synopsis image: Please provide a visual abstract as a high-resolution jpeg file 550 px-wide x 300-600 pixels high to illustrate your article.

9) As part of the EMBO Publications transparent editorial process initiative (see our Editorial at

<http://embomolmed.embopress.org/content/2/9/329>), EMBO Molecular Medicine will publish online a Review Process File (RPF) to accompany accepted manuscripts. This file will be published in conjunction with your paper and will include the anonymous referee reports, your point-by-point response and all pertinent correspondence relating to the manuscript. Let us know whether

you agree with the publication of the RPF and as here, if you want to remove or not any figures from it prior to publication. Please note that the Authors checklist will be published at the end of the RPF.

10) Please provide a point-by-point letter INCLUDING my comments as well as the reviewer's reports and your detailed responses (as Word file).

I look forward to reading a new revised version of your manuscript as soon as possible.

Yours sincerely,

Zeljko Durdevic

Zeljko Durdevic
Senior Editor
EMBO Molecular Medicine

*** Instructions to submit your revised manuscript ***

1) a .docx formatted version of the manuscript text (including Figure legends and tables)

2) Separate figure files*

3) supplemental information as Expanded View and/or Appendix. Please carefully check the authors guidelines for formatting Expanded view and Appendix figures and tables at <https://www.embopress.org/page/journal/17574684/authorguide#expandedview>

4) a letter INCLUDING the reviewer's reports and your detailed responses to their comments (as Word file).

5) The paper explained: EMBO Molecular Medicine articles are accompanied by a summary of the articles to emphasize the major findings in the paper and their medical implications for the non-specialist reader. Please provide a draft summary of your article highlighting

6) Author contributions: the contribution of every author must be detailed in a separate section.

7) EMBO Molecular Medicine now requires a complete author checklist (<https://www.embopress.org/page/journal/17574684/authorguide>) to be submitted with all revised manuscripts. Please use the checklist as guideline for the sort of information we need WITHIN the manuscript. The checklist should only be filled with page numbers where the information can be found. This is particularly important for animal reporting, antibody dilutions (missing) and exact values and n that should be indicated instead of a range.

8) Every published paper now includes a 'Synopsis' to further enhance discoverability. Synopses are displayed on the journal webpage and are freely accessible to all readers. They include a short stand first (maximum of 300 characters, including space) as well as 2-5 one sentence bullet points that summarise the paper. Please write the bullet points to summarise the key NEW findings. They should be designed to be complementary to the abstract - i.e. not repeat the same text. We encourage inclusion of key acronyms and quantitative information (maximum of 30 words / bullet point). Please use the passive voice. Please attach these in a separate file or send them by email, we will incorporate them accordingly.

You are also welcome to suggest a striking image or visual abstract to illustrate your article. If you do please provide a jpeg file 550 px-wide x 300-600px high.

9) A Conflict of Interest statement should be provided in the main text

10) Please note that we now mandate that all corresponding authors list an ORCID digital identifier. This takes <90 seconds to complete. We encourage all authors to supply an ORCID identifier, which will be linked to their name for unambiguous name identification.

Currently, our records indicate that the ORCID for your account is 0000-0002-8089-3237.

Link Not Available

11) Include a Reagents and Tools Table as part of the Methods section, which can be downloaded from our author guidelines (<https://www.embopress.org/page/journal/17574684/authorguide#structuredmethods>)

Photos 400-800 DPI

*Additional important information regarding figures and illustrations can be found at

<https://bit.ly/EMBOPressFigurePreparationGuideline>. See also figure legend preparation guidelines:

<https://www.embopress.org/page/journal/17574684/authorguide#figureformat>

The authors addressed the remaining editorial issues.

25th Jul 2025

Dear Dr. Pak,

We are pleased to inform you that your manuscript is accepted for publication and is now being sent to our publisher to be included in the next available issue of EMBO Molecular Medicine. Please be aware that datasets deposited in public repositories should be freely available upon publication.

Zeljko Durdevic
Senior Editor
EMBO Molecular Medicine
